# Deconstructing Self-Supervised Monocular Reconstruction: The Design Decisions that Matter

**Jaime Spencer**   *j.spencermartin@surrey.ac.uk*
*CVSSP, University of Surrey*

**Chris Russell**   *cmruss@amazon.com*
*Amazon*

**Simon Hadfield**   *s.hadfield@surrey.ac.uk*
*CVSSP, University of Surrey*

**Richard Bowden**   *r.bowden@surrey.ac.uk*
*CVSSP, University of Surrey*

**Reviewed on OpenReview:** *https://openreview.net/forum?id=GFK1FheE7F*

## Abstract

This paper presents an open and comprehensive framework to systematically evaluate state-of-the-art contributions to self-supervised monocular depth estimation. This includes pre-training, backbone, architectural design choices and loss functions. Many papers in this field claim novelty in either architecture design or loss formulation. However, simply updating the backbone of historical systems results in relative improvements of 25%, allowing them to outperform most modern systems. A systematic evaluation of papers in this field was not straightforward. The need to compare like-with-like in previous papers means that longstanding errors in the evaluation protocol are ubiquitous in the field. It is likely that many papers were not only optimized for particular datasets, but also for errors in the data and evaluation criteria. To aid future research in this area, we release a modular codebase (`https://github.com/jspenmar/monodepth_benchmark`), allowing for easy evaluation of alternate design decisions against corrected data and evaluation criteria. We re-implement, validate and re-evaluate 16 state-of-the-art contributions and introduce a new dataset (SYNS-Patches) containing dense outdoor depth maps in a variety of both natural and urban scenes. This allows for the computation of informative metrics in complex regions such as depth boundaries.

## 1 Introduction

Depth estimation is a fundamental low-level computer vision task that allows us to estimate the 3-D world from its 2-D projection(s). It is a core component enabling mid-level tasks such as SLAM, visual localization or object detection. More recently, it has heavily impacted fields such as Augmented Reality, autonomous vehicles and robotics, as knowing the real-world geometry of a scene is crucial for interacting with it—both virtually and physically.

This interest has resulted in a large influx of researchers hoping to contribute to the field and compare with previous approaches. New authors are faced with a complex range of established design decisions, such as the choice of backbone architecture & pretraining, loss functions and regularization. This is further complicated by the fact that papers are written to be accepted for publication. As such, they often emphasize the theoretical novelty of their work, over the robust design decisions that have the most impact on performance.

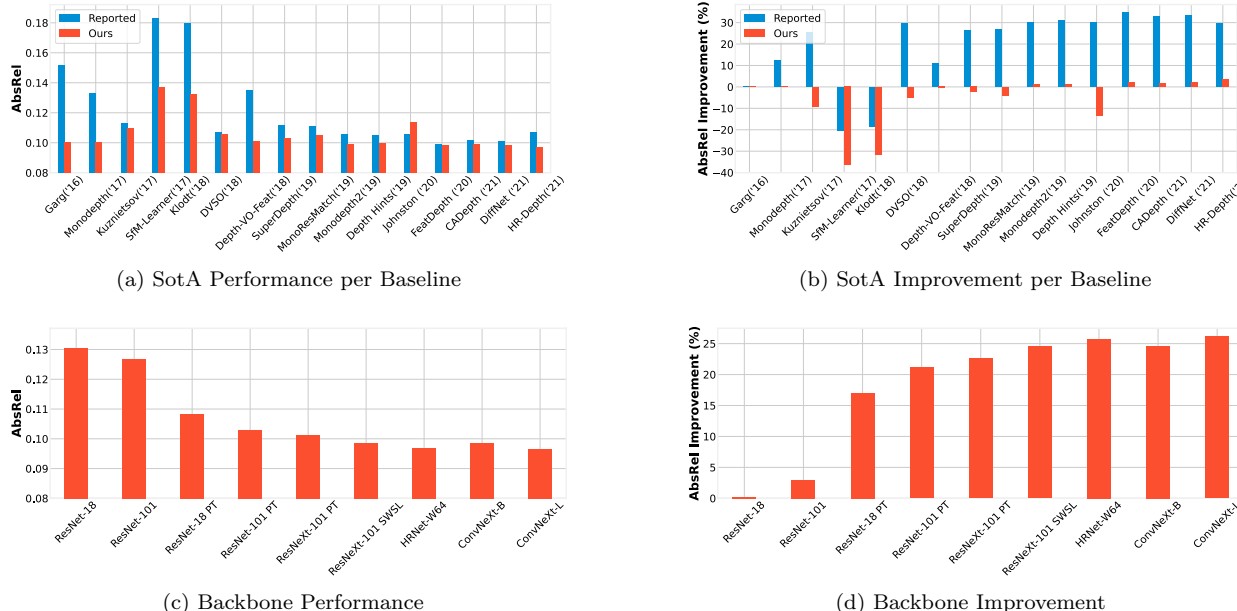

Figure 1: **Quantifying SotA Contributions. (a)** Performance at the time *vs.* our re-implementation using common design decisions (lower is better). Kuznietsov et al. (2017) and SfM-Learner (Zhou et al., 2017) have much higher error as they do not use stereo data training. **(b)** Original papers AbsRel relative performance improvement w.r.t. Garg et al. (2016) *vs.* real improvements observed training/evaluating baselines in a fair and comparable manner (higher is better). **(c)** Performance obtained by ablating the backbone architecture. **(d)** Backbone relative performance improvements (w.r.t. ResNet-18 from scratch) outweigh those provided by most recent contributions. See section Section 4.2 for more details of design choices.

This paper offers a chance to step back and re-evaluate the state of self-supervised monocular depth estimation. We do this via an extensive baseline study by carefully re-implementing popular State-of-the-Art (SotA) algorithms from scratch[1]. Our modular codebase allows us to study the impact of each framework component and ensure all approaches are trained in a fair and comparable way.

Figure 1 compares the performance reported by recent SotA against that obtained by the same technique on our updated benchmark. Our reimplementation improves performance for all evaluated baselines. However, the relative improvement resulting from each contribution is significantly lower than that reported by the original publications. In many cases, it is likely this is the result of arguably unfair comparisons against outdated baselines. For instance, as seen in Figures 1c & 1d, simply modernizing the choice of backbone in these legacy formulations results in performance gains of 25%. When applying these common design decisions to all approaches, it appears that 'legacy' formulations are still capable of outperforming many recent methods.

As part of our unified benchmark, we propose a novel evaluation dataset in addition to the exclusively used urban driving datasets. This new dataset (SYNS-Patches) contains 1175 images from a wide variety of urban and natural scenes, including categories such as urban residential, woodlands, indoor, industrial and more. This allows us to evaluate the generality of the learned depth models beyond the restricted automotive domain that is the focus of most papers. To summarize, the contributions of this paper are:

1. We provide a modular codebase containing modernized baselines that are easy to train and extend. This encourages direct like-with-like comparisons and better research practices.

2. We re-evaluate the updated baselines algorithms consistently using higher-quality corrected ground-truth on the existing benchmark dataset. This pushes the field away from commonly used flawed benchmarks, where errors are perpetuated for the sake of compatibility.

---

[1]Code is publicly available at `https://github.com/jspenmar/monodepth_benchmark`.

3. In addition to democratizing code and evaluation on the common Kitti benchmarks, we propose a novel testing dataset (SYNS-Patches) containing both urban and natural scenes. This focuses on the ability to generalize to a wider range of applications. The dense nature of the ground-truth allows us to provide informative metrics in complex regions such as depth boundaries.

4. We make these resources available to the wider research community, contributing to the further advancement of self-supervised monocular depth estimation.

## 2 Related Work

We consider self-supervised approaches that do not use ground-truth depth data at training time, but instead learn to predict depth as a way to estimate high-fidelity warps from one image to another. While all approaches predict depth from a single image, they can be categorized based on the additional frames used to perform these warps. Stereo-supervised frameworks directly predict metric depth given a known stereo baseline. Approaches using only monocular video additionally need to estimate Visual Odometry (VO) and only predict depth up to an unknown scale. These depth predictions are scaled during evaluation and aligned with the ground-truth. However, monocular methods are more flexible, since they do not require a stereo pair. Despite having their own artifacts, they do not share stereo occlusion artifacts, making video a valuable cue complementary to stereo.

### 2.1 Stereo

Garg et al. (2016) introduced view synthesis as a proxy task for self-supervised monocular depth estimation. The predicted depth map was used to synthesize the target view from its stereo pair, and optimized using an $L_1$ reconstruction loss. An additional smoothness regularization penalized all gradients in the predicted depth. Monodepth (Godard et al., 2017) used Spatial Transformer Networks (Jaderberg et al., 2015) to perform view synthesis in a fully-differentiable manner. The reconstruction loss was improved via a weighted $L_1$ and SSIM (Wang et al., 2004) photometric loss, while the smoothness regularization was softened in regions with strong image gradients. Monodepth additionally introduced a virtual stereo consistency term, forcing the network to predict both left and right depths from a single image. 3Net (Poggi et al., 2018) extended Monodepth to a trinocular setting by adding an extra decoder and treating the input as the central image in a three-camera rig. SuperDepth (Pillai et al., 2019) replaced Upsample-Conv blocks with sub-pixel convolutions (Shi et al., 2016), resulting in improvements when training with high-resolution images. They additionally introduced a differentiable stereo blending procedure based on test-time stereo blending (Godard et al., 2017). FAL-Net (Gonzalez Bello & Kim, 2020) proposed a discrete disparity volume network, complemented by a probabilistic view synthesis module and an occlusion-aware reconstruction loss.

Other methods complemented the self-supervised loss with (proxy) ground-truth depth regression. Kuznietsov et al. (2017) introduced a reverse Huber (berHu) regression loss (Zwald & Lambert-Lacroix, 2012; Laina et al., 2016) using the ground-truth sparse Light Detection and Ranging (LiDAR). Meanwhile, SVSM (Luo et al., 2018) proposed a two-stage pipeline in which a virtual stereo view was first synthesized from a self-supervised depth network. The target image and synthesized view were then processed in a stereo matching cost volume trained on the synthetic FlyingThings3D dataset (Mayer et al., 2016). Similarly, DVSO (Rui et al., 2018) and MonoResMatch (Tosi et al., 2019) incorporated stereo matching refinement networks to predict a residual disparity. These approaches used proxy ground-truth depth maps from direct stereo Simultaneous Localization and Mapping (SLAM) (Wang et al., 2017) and hand-crafted stereo matching (Hirschmüller, 2005), respectively. DVSO (Rui et al., 2018) introduced an occlusion regularization, encouraging sharper predictions that prefer background depths and second-order disparity smoothness. DepthHints (Watson et al., 2019) introduced proxy depth supervision into Monodepth2 (Godard et al., 2019), also obtained using SGM (Hirschmüller, 2005). The proxy depth maps were computed as the fused minimum reconstruction loss from predictions with various hyperparameters. As with the automasking procedure of Monodepth2 (Godard et al., 2019), the proxy regression loss was only applied to pixels where the hint produced a lower photometric reconstruction loss. Finally, PLADE-Net (Gonzalez Bello & Kim, 2021) expanded FAL-Net (Gonzalez Bello & Kim, 2020) by incorporating positional encoding and proxy depth regression using a matted Laplacian.

## 2.2 Monocular

SfM-Learner (Zhou et al., 2017) introduced the first approach supervised only by a stream of monocular images. This replaced the known stereo baseline with an additional network to regress VO between consecutive frames. An explainability mask was predicted to reduce the effect of incorrect correspondences from dynamic objects and occlusions. Klodt & Vedaldi (2018) introduced uncertainty (Kendall & Gal, 2017) alongside proxy SLAM supervision, allowing the network to ignore incorrect predictions. This uncertainty formulation also replaced the explainability mask from SfM-Learner. DDVO (Wang et al., 2018) introduced a differentiable DSO module (Engel et al., 2018) to refine the VO network prediction. They further made the observation that the commonly used edge-aware smoothness regularization (Godard et al., 2017) suffers from a degenerate solution in a monocular framework. This was accounted for by applying spatial normalization prior to regularizing.

Subsequent approaches focused on improving the robustness of the photometric loss. Monodepth2 (Godard et al., 2019) introduced several simple changes to explicitly address the assumptions made by the view synthesis framework. This included minimum reconstruction filtering to reduce occlusion artifacts, alongside static pixel automasking via the raw reconstruction loss. D3VO (Yang et al., 2020) additionally predicted affine brightness transformation parameters (Engel et al., 2018) for each support frame. Meanwhile, Depth-VO-Feat (Zhan et al., 2018) observed that the photometric loss was not always reliable due to ambiguous matching. They introduced an additional feature-based reconstruction loss synthesized from a pretrained dense feature representation (Weerasekera et al., 2017). DeFeat-Net (Spencer et al., 2020) extended this concept by learning dense features alongside depth, improving the robustness to adverse weather conditions and low-light environments. Shu et al. (2020) instead trained an autoencoder regularized to learn discriminative features with smooth second-order gradients.

Mahjourian et al. (2018) incorporated explicit geometric constraints via Iterative Closest Points using the predicted alignment pose and the mean residual distance. Since this process was non-differentiable, the gradients were approximated. SC-SfM-Learner (Bian et al., 2019) proposed an end-to-end differentiable geometric consistency constraint by synthesizing the support depth view. They included a variant of the absolute relative loss constrained to the range $[0, 1]$, additionally used as automasking for the reconstruction loss. Poggi et al. (2020) compared various approaches for estimating uncertainty in the depth prediction, including dropout, ensembles, student-teacher training and more. Johnston & Carneiro (2020) proposed to use a discrete disparity volume and the variance along each pixel to estimate the uncertainty of the prediction. During training, the view reconstruction loss was computed using the Expected disparity, *i.e.* the weighted sum based on the likelihood of each bin.

Recent approaches have focused on developing architectures to produce higher resolution predictions that do not suffer from interpolation artifacts. PackNet (Guizilini et al., 2020) proposed an encoder-decoder network using 3-D (un)packing blocks with sub-pixel convolutions (Shi et al., 2016). This allowed the network to encode spatial information in an invertible manner, used by the decoder to make higher quality predictions. However, this came at the cost of a tenfold increase in parameters. CADepth (Yan et al., 2021) proposed a Structure Perception self-attention block as the final encoder stage, providing additional context to the decoder. This was complemented by a Detail Emphasis module, which refined skip connections using channel-wise attention. Meanwhile, Zhou et al. (2021) replaced the commonly used ResNet encoder with HRNet due to its suitability for dense predictions. Similarly, concatenation skip connections were replaced with a channel/spatial attention block. HR-Depth (Lyu et al., 2021) introduced a highly efficient decoder based on SqueezeExcitation blocks (Hu et al., 2020) and progressive skip connections.

While each of these methods reports improvements over previous approaches, they are frequently not directly comparable. Most approaches differ in the number of training epochs, pretraining datasets, backbone architectures, post-processing, image resolutions and more. This begs the question as to what percentage of the improvements are due to the fundamental contributions of each approach, and how much is unaccounted for in the silent changes. In this paper, we aim to answer this question by first studying the effect of changing components rarely claimed as contributions. Based on the findings, we train recent SotA methods in a comparable way, further improving their performance and evaluating each contribution independently.

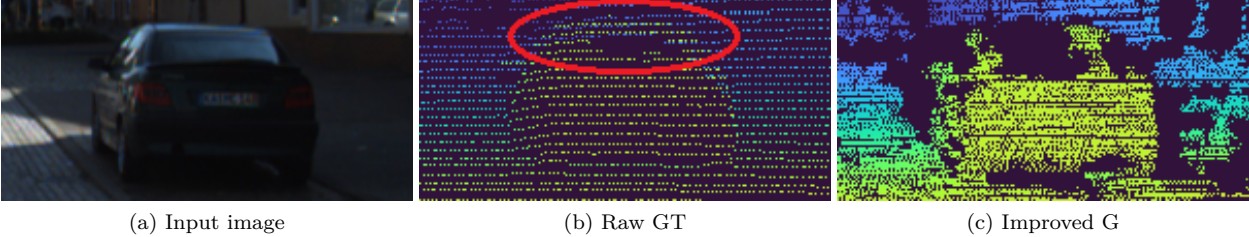

(a) Input image          (b) Raw GT          (c) Improved G

Figure 2: **Inaccurate Ground-Truth.** The original Kitti (Geiger et al., 2013) ground-truth data used by previous monocular depth benchmarks (Eigen & Fergus, 2015) is inaccurate and contains errors, especially at object boundaries. Note the background bleeding in the highlighted region. Uhrig et al. (2018) correct this by accumulating LiDAR data over multiple frames.

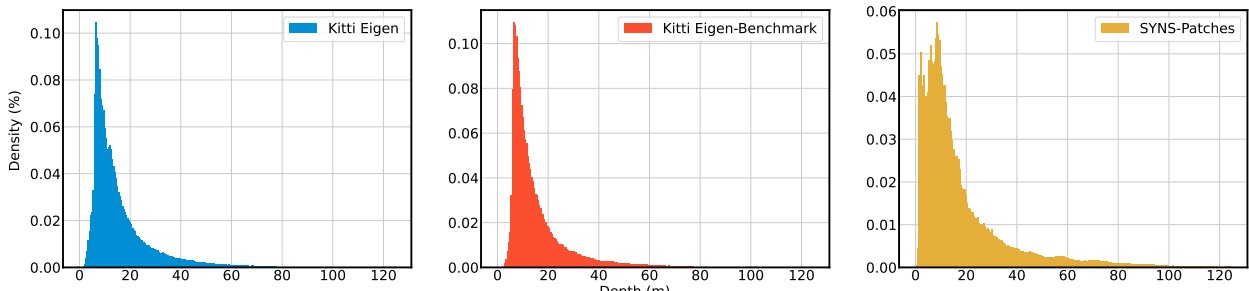

Figure 3: **Depth Distribution.** We show the distribution of depths for KE, KEB & SYNS-Patches. SYNS-Patches contains more varied depth values due to the indoors scenes. In all cases the maximum depth is clamped to 100 meters during evaluation.

## 3 Benchmark Datasets

The objective of this paper is to critically examine recent SotA contributions to monocular depth learning. One of the biggest hurdles to overcome is the lack of informative benchmarks due to erroneous evaluation procedures. By far, the most common evaluation dataset in the field is the Kitti Eigen (KE) split (Eigen & Fergus, 2015). Unfortunately, it has contained critical errors since its creation, the most egregious being the inaccuracy of the ground-truth depth maps. The original Kitti data suffered from artifacts due to the camera and LiDAR not being identically positioned. As such, each sensor had slightly different viewpoints, each with slightly different occlusions. This was exacerbated by the sparsity of the LiDAR, resulting in the background bleeding into the foreground. An example of this can be seen in Figure 2. Furthermore, the conversion to depth maps omitted the transformation to the camera reference frame and used the raw LiDAR depth instead. Finally, the Squared Relative error was computed incorrectly without the squared term in the denominator.

Although these errors have been noted by previous works, they have nevertheless been propagated from method to method up to this day due to the need to compare like-with-like when reporting results. It is much easier to simply ignore these errors and copy the results from existing papers, than it is to correct the evaluation procedure and re-evaluate previous baselines. We argue this behavior should be corrected immediately and provide an open-source codebase to make the transition simple for future authors. Our benchmark consists of two datasets: the Kitti Eigen-Benchmark (KEB) split (Uhrig et al., 2018) and the newly introduced SYNS-Patches dataset.

### 3.1 Kitti Eigen-Benchmark

Uhrig et al. (2018) aimed to fix the aforementioned errors in the Kitti (Geiger et al., 2013) dataset. This was done by accumulating LiDAR data over ±5 frames to create denser ground-truth and removing occlusion artifacts via SGM (Hirschmüller, 2005) consistency checks. The final KEB split represents the subset of KE with available corrected ground-truth depth maps. This consists of 652 of the original 697 images.

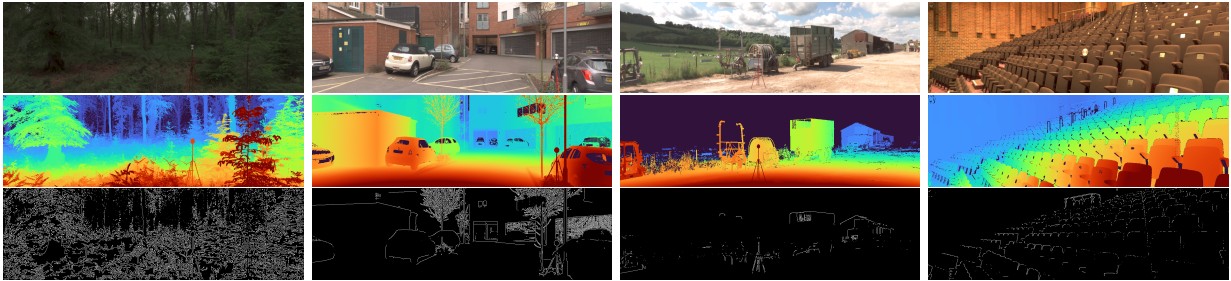

Figure 4: **SYNS-Patches.** Top: Diverse testing images. Middle: Dense depth maps. Bottom: Log-depth Canny edges.

Table 1: **SYNS-Patches Scenes.** We show the distibution of images per scene in the proposed dataset. This evaluates the model's capability to generalize beyond purely automotive data.

| Agriculture | Natural | Indoor | Woodland | Residential | Industry | Misc | Recreation | Transport | Total |
|---|---|---|---|---|---|---|---|---|---|
| 315 | 183 | 148 | 144 | 123 | 107 | 72 | 62 | 21 | 1,175 |

Similarly, we report the well-established image-based metrics from the official Kitti Benchmark. While some of these overlap with KE, we avoid saturated metrics such as $\delta < 1.25^3$ or the incorrect SqRel error[2]. We also report pointcloud-based reconstruction metrics proposed by Örnek et al. (2022). They argue that image-based depth metrics are insufficient to accurately evaluate depth estimation models, since the true objective is to reconstruct the 3-D scene. Further details can be found in Section B.1. Nevertheless, the Kitti dataset is becoming saturated and a more varied and complex evaluation framework is required.

## 3.2 SYNS-Patches

The second dataset in our benchmark is the novel SYNS-Patches, based on SYNS (Adams et al., 2016). The original SYNS is composed of 92 scenes from 9 different categories. Each scene contains a panoramic HDR image and an aligned dense LiDAR scan. This provides a previously unseen dataset to evaluate the generalization capabilities of the trained baselines. We extend depth estimation to a wider variety of environments, including woodland & natural scenes, industrial estates, indoor scenes and more. SYNS provides dense outdoor LiDAR scans. Previous dense depth datasets (Koch et al., 2018) are typically limited to indoor scenes, while outdoor datasets (Geiger et al., 2013; Guizilini et al., 2020) are sparse. These dense depth maps allow us to compute metrics targeting high-interest regions such as depth boundaries.

Our dataset is generated by sampling 18 undistorted patches per scene, performing a full horizontal rotation roughly at eye level. To make this dataset more amenable to the transition from Kitti, we maintain the same aspect ratio and extract patches of size $376 \times 1242$. We follow the same procedure on the LiDAR to extract aligned dense ground-truth depth maps. Ground-truth depth boundaries are obtained via Canny edges in the dense log-depth map. After manual validation and removal of data with dynamic object artifacts, the final test set contains 1,175 of the possible 1,656 images. We show the distributions of depth values in Figure 3, the images per category in Table 1 and some illustrative examples in Figure 4. For each dataset, we additionally compute the percentage of image pixels with ground-truth depth values. The original Kitti Eigen has a density of only 4.10%, while the improved Kitti Eigen-Benchmark has 15.28%. Meanwhile, SYNS-Patches has ground-truth for 78.30% of the scene, demonstrating the high-quality of the data.

We report image-based metrics from Uhrig et al. (2018) and pointcloud-based metrics from Örnek et al. (2022). For more granular results, we compute these metrics only at depth boundary pixels. Finally, we compute the edge-based accuracy and completeness from Koch et al. (2018) using the Chamfer pixel distance to/from predicted and ground-truth depth edges. Further dataset creation details can be found in Section B.3. Note that SYNS-Patches is purely a testing dataset never used for training. This represents completely unseen environments that test the generalization capabilities of the trained models.

---

[2]The squared term was missing from the denominator.

## 4 The Design Decisions That Matter

Most contributions to self-supervised monocular depth estimation focus on alterations to the view synthesis loss (Godard et al., 2017; 2019; Zhan et al., 2018), additional geometric consistency (Mahjourian et al., 2018; Bian et al., 2019) or regularization (Rui et al., 2018) and the introduction of proxy supervised losses (Kuznietsov et al., 2017; Tosi et al., 2019; Watson et al., 2019). In this paper, we return to first principles and study the effect of changing components in the framework rarely claimed as contributions. We focus on practical changes that lead to significant improvements, raising the overall baseline performance and providing a solid platform on which to evaluate recent SotA models.

Since this paper primarily focuses on the benchmarking and evaluation procedure, we provide a detailed review of monocular depth estimation and each contribution as supplementary material in Section A. We encourage readers new to the field to refer to this section for additional details. It is worth reiterating that the depth estimation network only ever requires a single image as its input, during both training and evaluation. However, methods that use monocular video sequences during training require an additional relative pose regression network. This replaces the known fixed stereo baseline used by stereo-trained models. Note that this pose network is only required during training to perform the view synthesis and compute the photometric loss. As such, it can be discarded during the depth evaluation. Furthermore, since all monocular-trained approaches use the same pose regression system, it is beyond the scope of this paper to evaluate the performance of this component.

### 4.1 Implementation details

We train these models on the common Kitti Eigen-Zhou (KEZ) training split (Zhou et al., 2017), containing 39,810 frames from the KE split where static frames are discarded. Most previous works perform their ablation studies on the KE test set, where the final models are also evaluated. This indirectly incorporates the testing data into the hyperparameter optimization cycle, which can lead to overfitting to the test set and exaggerated performance claims. We instead use a random set of 700 images from the KEZ validation split with updated ground-truth depth maps (Uhrig et al., 2018). Furthermore, we report the image-based metrics from the Kitti Benchmark and the pointcloud-based metrics proposed by Örnek et al. (2022), as detailed in Section B.2. For ease of comparison, we add the performance rank ordering of the various methods. Image-based ordering uses AbsRel, while pointcloud-based uses the F-Score.

Models were trained for 30 epochs using Adam with a base learning rate of $1e$-$4$, reduced to $1e$-$5$ halfway through training. The default DepthNet backbone is a pretrained ConvNeXt-T (Liu et al., 2022), while PoseNet uses a pretrained ResNet-18 (He et al., 2016). We use an image resolution of $192 \times 640$ with a batch size of 8. Horizontal flips and color jittering are randomly applied with a probability of 0.5.

We adopt the minimum reconstruction loss and static pixel automasking losses from Monodepth2 (Godard et al., 2019), due to their simplicity and effectiveness. We use edge-aware smoothness regularization (Godard et al., 2017) with a scaling factor of 0.001. These losses are computed across all decoder scales, with the intermediate predictions upsampled to match the full resolution. We train in a Mono+Stereo setting, using monocular video and stereo pair support frames. To account for the inherently random optimization procedure, each model variant is trained using three random seeds and mean performance is reported. We emphasize that the training code has been publicly released alongside the benchmark code to ensure the reproducibility of our results and to allow future researchers to build off them.

### 4.2 Backbone Architecture & Pretraining

Here we evaluate the performance of recent SotA backbone architectures and their choice of pretraining. Results can be found in Table 2 & Figure 5. We test ResNet (He et al., 2016), ResNeXt (Xie et al., 2017), MobileNet-v3 (Howard et al., 2019), EfficientNet (Tan & Le, 2019), HRNet (Wang et al., 2021) and ConvNeXt (Liu et al., 2022). ResNet variants are trained either from scratch or using pretrained supervised weights from Wightman (2019). ResNeXt-101 variants additionally include fully supervised (Wightman, 2019), weakly-supervised and self-supervised (Yalniz et al., 2019) weights. All remaining backbones are pretrained by default.

Table 2: **Backbone Ablation.** We study the effect of various backbone architectures & pretraining methods. PT indicates use of pretrained ResNet weights. All other baselines are pretrained by default. ConvNeXt provides the best performance, followed by HRNet-W64. ResNeXt can be further improved by using self-/weakly-supervised weights. Frames per second were measured on an NVIDIA GeForce RTX 3090 with an image of size 192 × 640.

| *KEZ (val)* | MParam↓ | FPS↑ | # | MAE↓ | RMSE↓ | AbsRel↓ | LogSI↓ | # | Chamfer↓ | F-Score↑ | IoU↑ |
|---|---|---|---|---|---|---|---|---|---|---|---|
| | | | | | Image-based | | | | | Pointcloud-based | |
| ResNet-18 | 14.33 | 237.1 | 16 | 1.50 | 3.58 | 7.70 | 11.45 | 16 | 0.59 | 50.71 | 35.20 |
| ResNet-18 PT | 14.33 | 238.2 | 10 | 1.23 | 3.06 | 6.29 | 9.03 | 10 | 0.49 | 54.35 | 38.79 |
| ResNet-101 | 51.51 | 79.63 | 13 | 1.34 | 3.27 | 6.76 | 10.33 | 12 | 0.53 | 53.53 | 37.80 |
| ResNet-101 PT | 51.51 | 79.72 | 7 | 1.16 | 2.95 | 5.71 | 8.94 | 4 | 0.47 | 57.25 | 41.35 |
| ResNeXt-101 | 95.76 | 72.07 | 11 | 1.29 | 3.17 | 6.55 | 9.74 | 11 | 0.51 | 53.80 | 38.16 |
| ResNeXt-101 PT | 95.76 | 71.98 | 8 | 1.09 | 2.64 | 5.74 | 7.77 | 9 | 0.44 | 55.10 | 39.55 |
| ResNeXt-101 SSL | 95.76 | 71.50 | 6 | 1.08 | 2.62 | 5.65 | 7.74 | 7 | 0.44 | 55.56 | 39.96 |
| ResNeXt-101 SWSL | 95.76 | 71.77 | 5 | 1.07 | 2.65 | 5.56 | 7.78 | 6 | 0.44 | 56.40 | 40.83 |
| MobileNet-v3-S | 2.20 | 158.3 | 17 | 1.95 | 4.27 | 10.25 | 14.74 | 17 | 0.76 | 45.13 | 30.35 |
| MobileNet-v3-L | 6.67 | 137.8 | 12 | 1.27 | 3.05 | 6.72 | 9.00 | 13 | 0.51 | 52.87 | 37.46 |
| EfficientNet-B0 | 5.84 | 99.35 | 14 | 1.29 | 3.04 | 6.95 | 8.90 | 14 | 0.52 | 52.14 | 37.05 |
| EfficientNet-B4 | 19.41 | 54.28 | 15 | 1.33 | 3.11 | 7.20 | 9.22 | 15 | 0.53 | 51.77 | 36.83 |
| HRNet-W18 | 16.05 | 29.65 | 9 | 1.14 | 2.81 | 5.86 | 8.17 | 8 | 0.47 | 55.34 | 39.76 |
| HRNet-W64 | 122.81 | 24.17 | 4 | 1.02 | 2.51 | 5.33 | _7.26_ | 5 | 0.43 | 57.05 | 41.45 |
| ConvNeXt-T | 31.93 | 147.5 | 3 | 1.03 | 2.59 | 5.30 | 7.63 | 3 | 0.43 | 57.97 | 42.26 |
| ConvNeXt-B | 92.65 | 98.09 | **1** | **0.97** | _2.49_ | **4.98** | 7.40 | **1** | **0.40** | **60.63** | **44.89** |
| ConvNeXt-L | 203.27 | 72.44 | _2_ | _0.98_ | **2.44** | _5.19_ | **7.07** | _2_ | _0.41_ | _58.17_ | _42.49_ |

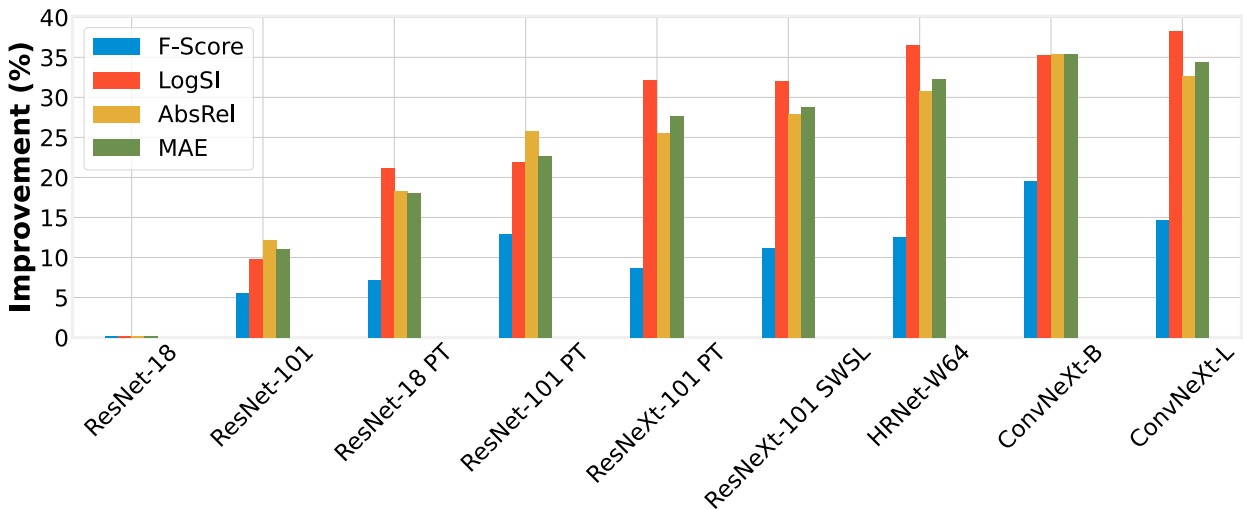

Figure 5: **Backbone Ablation.** We show the relative performance improvement in F-Score, MAE, LogSI and AbsRel obtained by different backbone architectures and pretraining methods. Most existing papers limit their backbone to a pretrained ResNet-18, resulting in limited improvements. Full results in Table 2.

As seen in Table 2, ConvNeXt variants outperform all other backbones, with HRNet-W64 following closely behind. Within lighter backbones, *i.e.* < 20 MParams, we find HRNet-W18 to be the most effective, greatly outperforming mobile backbones such as MobileNet-v3 and EfficientNet. Regarding pretraining, all ResNet backbones show significant improvements when using pretrained weights. ResNeXt variants are further improved by using pretrained weights from self-supervised or weakly-supervised training (Yalniz et al., 2019).

A summary of these results can be found in Figure 5, showing the relative improvement over ResNet-18 (from scratch) in F-Score, MAE, LogSI and AbsRel metrics. The vast majority of monocular depth approaches simply stop at a pretrained ResNet-18 backbone (∼20% improvement). However, replacing this with a well-engineered modern architecture such as ConvNeXt or HRNet results in an additional 15% improvement. As such, the remainder of the paper will use the ConvNeXt-B backbone when comparing baselines.

### 4.3 Depth Regularization

The second study evaluates the performance of various commonly used depth regularization losses. We focus on variants of depth spatial gradient smoothing such as first-order (Garg et al., 2016), edge-aware (Godard et al., 2017), second-order (Rui et al., 2018) and Gaussian blurring. We additionally test two variants of occlusion regularization (Rui et al., 2018), favoring either background or foreground depths.

Results are shown in Table 3. We evaluate these models on the proposed KEZ validation split, as well as the KE test set commonly used by previous papers. Once again, the KEZ split is used to limit the chance of overfitting to the target KEB and SYNS-Patches test sets. When evaluating on the updated ground-truth, using no additional regularization produces the best results. All variants of depth smoothness produce slightly inferior results, all comparable to each other. Incorporating occlusion regularization alongside smoothness regularization again leads to a decrease in performance. Meanwhile, on the inaccurate ground-truth (Eigen & Fergus, 2015), smoothness constraints produce slightly better results. We believe this is due to the regularization encouraging oversmoothing in boundaries, which mitigates the effect of the incorrect ground-truth boundaries shown in Figure 2. As such, this is overfitting to errors in the evaluation criteria, rather than improving the actual depth prediction. Meanwhile, the large overlapping receptive fields of modern architectures such as ConvNeXt are capable of implicitly providing the smoothness required by neighbouring depths. Once again, these results point towards the need for an up-to-date benchmarking procedure that is based on reliable data and more informative metrics.

Table 3: **Depth Regularization Ablation.** We study the effect of adding smoothness (edge-aware, first-/second-order, w/wo Gaussian blurring) and occlusion regularization (prefer background/foreground). Whilst all methods typically use these regularizations, omitting them provides the best performance when evaluating on the corrected ground-truth. Meanwhile, these regularizations provide slight improvements on the outdated Kitti Eigen split. This is likely due to oversmoothing to account for the inaccurate boundaries shown in Figure 2.

| | | Image-based | | | | | Pointcloud-based | | |
|---|---|---|---|---|---|---|---|---|---|
| *KEZ (val)* | # | MAE↓ | RMSE↓ | AbsRel↓ | LogSI↓ | # | Chamfer↓ | F-Score↑ | IoU↑ |
| No Regularization | **1** | **1.63** | 3.68 | **7.86** | 11.35 | **1** | **0.66** | **50.25** | **34.68** |
| First-order | 3 | 1.65 | **3.64** | 8.12 | 11.32 | 6 | 0.67 | 49.30 | 33.90 |
| Fist-order Blur | 5 | 1.65 | 3.66 | 8.19 | 11.36 | 4 | 0.67 | 49.39 | 33.96 |
| Second-order | 2 | 1.64 | 3.64 | 8.10 | 11.25 | 5 | 0.67 | 49.32 | 33.93 |
| Second-order Blur | 4 | 1.65 | 3.66 | 8.14 | 11.27 | 2 | 0.67 | 49.46 | 34.05 |
| Occlusion (BG) | 7 | 1.66 | 3.65 | 8.27 | 11.38 | 7 | 0.68 | 48.86 | 33.52 |
| Occlusion (FG) | 6 | 1.65 | 3.65 | 8.19 | **11.23** | 3 | 0.67 | 49.40 | 34.02 |

| *KE (test)* | # | AbsRel↓ | SqRel↓ | RMSE↓ | LogRMSE↓ | $\delta < 1.25^1$↑ | $\delta < 1.25^2$↑ | $\delta < 1.25^3$↑ |
|---|---|---|---|---|---|---|---|---|
| No Regularization | 2 | 0.1002 | 0.7580 | 4.5833 | 0.1905 | 0.8865 | 0.9589 | 0.9798 |
| First-order | **1** | **0.0993** | **0.7386** | 4.5274 | 0.1873 | 0.8870 | 0.9608 | 0.9810 |
| Fist-order Blur | 7 | 0.1013 | 0.7600 | 4.5431 | 0.1886 | 0.8849 | 0.9603 | 0.9807 |
| Second-order | 4 | 0.1004 | 0.7567 | **4.5260** | 0.1877 | **0.8871** | 0.9609 | 0.9808 |
| Second-order Blur | 3 | 0.1003 | 0.7661 | 4.5512 | 0.1881 | 0.8867 | 0.9607 | 0.9806 |
| Occlusion (BG) | 6 | 0.1004 | 0.7573 | 4.5454 | 0.1872 | 0.8850 | 0.9607 | **0.9811** |
| Occlusion (FG) | 5 | 0.1004 | 0.7623 | 4.5326 | **0.1872** | 0.8870 | **0.9611** | 0.9810 |

Table 4: **State-of-the-Art Summary.** We summarize the settings used by each evaluated method in the benchmark. Contributions made by each method are indicated by either bold font or an asterisk. Please note that these settings do not exactly reflect the original implementations by the respective authors, since we have introduced changes for the sake of comparability and performance improvement.

| | Train | Decoder | Proxy | Min+Mask | Feat | Virtual | Blend | Mask | Smooth | Occ |
|---|---|---|---|---|---|---|---|---|---|---|
| SfM-Learner | **M** | Monodepth | | | | | | **Explainability** | ✓ | |
| Klodt | M | Monodepth | | | | | | **Uncertainty** | ✓ | |
| Monodepth2 | M | Monodepth | | ✓* | | | | | ✓ | |
| Johnston | M | **Discrete** | | ✓ | | | | | ✓ | |
| HR-Depth | M | **HR-Depth** | | ✓ | | | | | ✓ | |
| Garg | **S** | Monodepth | | | | | | | ✓ | |
| Monodepth | S | **Monodepth** | | | | ✓* | | | ✓* | |
| SuperDepth | S | **Sub-Pixel** | | | | | ✓* | | ✓ | ✓ |
| Depth-VO-Feat | **MS** | Monodepth | | | **DepthNet** | | | | ✓ | |
| Monodepth2 | MS | Monodepth | | ✓* | | | | | ✓ | |
| FeatDepth | MS | Monodepth | | ✓ | **AutoEnc** | | | | ✓ | |
| CADepth | MS | **CADepth** | | ✓ | | | | | ✓ | |
| DiffNet | MS | **DiffNet** | | ✓ | | | | | ✓ | |
| HR-Depth | MS | **HR-Depth** | | ✓ | | | | | ✓ | |
| Kuznietsov | **SD*** | Monodepth | **berHu** | | | | | | ✓ | |
| DVSO | SD* | Monodepth | berHu | | | ✓ | | | ✓* | ✓* |
| MonoResMatch | SD* | Monodepth | berHu | | | ✓ | | | ✓ | |
| DepthHints | **MSD*** | Monodepth | **LogL1** | ✓ | | | | | ✓ | |
| Ours | MS | HR-Depth | | | | | | | | |
| Ours (Min) | MS | HR-Depth | | ✓ | | | | | | |
| Ours (Proxy) | MSD* | HR-Depth | LogL1 | | | | | | | |
| Ours (Min+Proxy) | MSD* | HR-Depth | LogL1 | ✓ | | | | | | |

## 5 Results

This section presents the results obtained when combining recent SotA approaches with our proposed design changes. Once again, the focus lies in training all baselines in a fair and comparable manner, with the aim of finding the effectiveness of each contribution. For completeness and comparison with the original papers, we report results on the KE split in Section 5.2. However, it is worth reiterating that *we strongly believe this evaluation should not be used by future authors.*

### 5.1 Evaluation details

We cap the maximum depth to 100 meters (compared to the common 50m (Garg et al., 2016) or 80m (Zhou et al., 2017)) and omit border cropping (Garg et al., 2016) & stereo-blending post-processing (Godard et al., 2017). Again, we show the rank ordering based on image-based (AbsRel), pointcloud-based (F-Score) and edge-based (F-Score) metrics. Stereo-supervised (**S** or **MS**) approaches apply a fixed scaling factor to the prediction, due to the known baseline between the cameras during training. Monocular-supervised (**M**) methods instead apply per-image median scaling to align the prediction and ground-truth.

We largely follow the implementation details outlined in Section 4.1, except for the backbone architecture, which is replaced with ConvNeXt-B (Liu et al., 2022). As a baseline, all methods use the standard DispNet decoder (Mayer et al., 2016) (excluding methods proposing new architectures), the SSIM+L1 photometric loss (Godard et al., 2017), edge-aware smoothness regularization (Godard et al., 2017) and upsampled multi-scale losses (Godard et al., 2019). We settle on these design decisions due to their popularity and prevalence in all recent approaches. This allows us to minimize the changes between legacy and modern approaches and focus on the contributions of each method.

The specific settings and contributions of each approach can be found in Table 4. For the full details please refer to the original publication, the review in Section 2 or the public codebase. We label the training supervision as follows: **M** = Monocular video synthesis, **S** = Stereo pair synthesis & **D\*** = Proxy depth regression. In this case, all proxy depth maps used for regression (**D\***) are obtained via the hand-crafted stereo disparity algorithm SGM (Hirschmüller, 2005). We further improve their robustness via the min reconstruction fusion proposed by Depth Hints (Watson et al., 2019).

Table 5: **Kitti Eigen Evaluation.** Results reported in the original publications (top) *vs.* those obtained by our updated baselines (bottom). The proposed baselines outperform those provided by the original authors in every case, most notably in the mono (Zhou et al., 2017) and stereo (Garg et al., 2016) baselines. For instance $\delta < 1.25$ accuracy is improved by a raw 10% in both cases, while AbsRel error is decreased by 5%. As such, models with the proposed changes represent the new SotA.

| *Original results* | Train | # | AbsRel↓ | SqRel↓ | RMSE↓ | LogRMSE↓ | $\delta < 1.25^1$↑ | $\delta < 1.25^2$↑ | $\delta < 1.25^3$↑ |
|---|---|---|---|---|---|---|---|---|---|
| SfM-Learner | M | 18 | 0.1830 | 1.5950 | 6.7090 | 0.2700 | 0.7340 | 0.9020 | 0.9590 |
| Klodt | M | 17 | 0.1800 | 1.9700 | 6.8550 | 0.2700 | 0.7650 | 0.9130 | 0.9620 |
| Monodepth2 | M | 13 | 0.1150 | 0.9030 | 4.8630 | 0.1930 | 0.8770 | 0.9590 | 0.9810 |
| Johnston | M | 6 | 0.1060 | 0.8610 | 4.6990 | 0.1850 | 0.8890 | 0.9620 | 0.9820 |
| HR-Depth | M | 9 | 0.1090 | 0.7920 | 4.6320 | 0.1850 | 0.8840 | 0.9620 | 0.9830 |
| Garg | S | 16 | 0.1520 | 1.2260 | 5.8490 | 0.2460 | 0.7840 | 0.9210 | 0.9670 |
| Monodepth | S | 14 | 0.1330 | 1.1420 | 5.5330 | 0.2300 | 0.8300 | 0.9360 | 0.9700 |
| SuperDepth | S | 11 | 0.1120 | 0.8750 | 4.9580 | 0.2070 | 0.8520 | 0.9470 | 0.9770 |
| Depth-VO-Feat | MS | 15 | 0.1350 | 1.1320 | 5.5850 | 0.2290 | 0.8200 | 0.9330 | 0.9710 |
| Monodepth2 | MS | 5 | 0.1060 | 0.8180 | 4.7500 | 0.1960 | 0.8740 | 0.9570 | 0.9790 |
| FeatDepth | MS | **1** | **0.0990** | **0.6970** | **4.4270** | 0.1840 | 0.8890 | 0.9630 | 0.9820 |
| CADepth | MS | 3 | 0.1020 | 0.7520 | 4.5040 | 0.1810 | 0.8940 | 0.9640 | 0.9830 |
| DiffNet | MS | 2 | 0.1010 | 0.7490 | 4.4450 | **0.1790** | **0.8980** | **0.9650** | 0.9830 |
| HR-Depth | MS | 7 | 0.1070 | 0.7850 | 4.6120 | 0.1850 | 0.8870 | 0.9620 | 0.9820 |
| Kuznietsov | SD* | 12 | 0.1130 | 0.7410 | 4.6210 | 0.1890 | 0.8620 | 0.9600 | **0.9860** |
| DVSO | SD* | 8 | 0.1070 | 0.8520 | 4.7850 | 0.1990 | 0.8660 | 0.9500 | 0.9780 |
| MonoResMatch | SD* | 10 | 0.1110 | 0.8670 | 4.7140 | 0.1990 | 0.8640 | 0.9540 | 0.9790 |
| DepthHints | MSD* | 4 | 0.1050 | 0.7690 | 4.6270 | 0.1890 | 0.8750 | 0.9590 | 0.9820 |

| *Our implementation* | Train | # | AbsRel↓ | SqRel↓ | RMSE↓ | LogRMSE↓ | $\delta < 1.25^1$↑ | $\delta < 1.25^2$↑ | $\delta < 1.25^3$↑ |
|---|---|---|---|---|---|---|---|---|---|
| SfM-Learner | M | 18 | 0.1374 | 1.6426 | 5.3609 | 0.2201 | 0.8562 | 0.9467 | 0.9732 |
| Klodt | M | 17 | 0.1325 | 1.4591 | 5.2738 | 0.2178 | 0.8604 | 0.9482 | 0.9740 |
| Monodepth2 | M | 16 | 0.1145 | 0.9374 | 4.9131 | 0.1946 | 0.8767 | 0.9589 | 0.9803 |
| Johnston | M | 15 | 0.1140 | 0.8577 | 4.7707 | 0.1912 | 0.8772 | 0.9606 | 0.9816 |
| HR-Depth | M | 14 | 0.1114 | 0.9026 | 4.8276 | 0.1911 | 0.8817 | 0.9609 | 0.9815 |
| Garg | S | 8 | 0.1007 | 0.8180 | 4.6693 | 0.1922 | 0.8859 | 0.9575 | 0.9788 |
| Monodepth | S | 7 | 0.1003 | 0.7958 | 4.6448 | 0.1909 | 0.8846 | 0.9573 | 0.9793 |
| SuperDepth | S | 10 | 0.1030 | 0.8122 | 4.6907 | 0.1939 | 0.8828 | 0.9562 | 0.9786 |
| Depth-VO-Feat | MS | 9 | 0.1012 | 0.7802 | 4.6676 | 0.1954 | 0.8800 | 0.9551 | 0.9780 |
| Monodepth2 | MS | 5 | 0.0994 | 0.7491 | 4.5438 | 0.1879 | 0.8883 | 0.9607 | 0.9806 |
| FeatDepth | MS | 3 | 0.0986 | 0.7410 | 4.5521 | 0.1880 | 0.8868 | 0.9603 | 0.9805 |
| CADepth | MS | 4 | 0.0988 | 0.7313 | 4.5117 | 0.1848 | 0.8876 | 0.9621 | 0.9816 |
| DiffNet | MS | 2 | 0.0983 | 0.7377 | 4.5139 | 0.1852 | 0.8900 | 0.9622 | 0.9814 |
| HR-Depth | MS | **1** | **0.0974** | **0.7306** | **4.4875** | 0.1854 | 0.8921 | 0.9621 | 0.9811 |
| Kuznietsov | SD* | 13 | 0.1098 | 0.9175 | 4.7278 | 0.1846 | 0.8797 | 0.9625 | **0.9832** |
| DVSO | SD* | 12 | 0.1056 | 0.8226 | 4.6427 | 0.1880 | 0.8822 | 0.9594 | 0.9812 |
| MonoResMatch | SD* | 11 | 0.1048 | 0.8266 | 4.6263 | 0.1881 | 0.8835 | 0.9588 | 0.9806 |
| DepthHints | MSD* | 6 | 0.0997 | 0.7958 | 4.5112 | **0.1834** | **0.8924** | **0.9631** | 0.9820 |

### 5.2 Kitti Eigen

We first validate the effectiveness of our improved implementations on the KE split. As discussed, we strongly believe that this dataset encourages suboptimal design decisions, and future work should not evaluate on it. We report the original metrics from Eigen & Fergus (2015), as detailed in Section B.1. Results can be found in Table 5, alongside results from the original published papers. As seen, the models trained with our design decisions improve over each of their respective baselines. This is particularly noticeable in the early baselines (Garg et al., 2016; Zhou et al., 2017), which have remained unchanged since publication. This again highlights the importance of providing up-to-date baselines that are trained in a comparable way.

### 5.3 Kitti Eigen-Benchmark

We report results on the KEB split, described in Sections 3.1 & B.2. To re-iterate, this is an updated evaluation using the corrected depth maps from Uhrig et al. (2018). From these images, we select a subset of 10 interesting images to evaluate the qualitative performance of the trained models. Results can be found in Table 6 & Figure 7. As shown, when training and evaluating in a comparable manner, the improvements provided by recent contributions are significantly lower than those reported in the original papers. In fact, we find the seminal stereo method by Garg et al. (2016) to be one of the top performers across all metrics. Most notably, it outperforms all other methods in 3-D pointcloud-based metrics, indicating that the reconstructions are the most accurate. Incorporating the discussed design decisions along with SotA contributions provides the best image-based performance. However, these contributions were not developed to optimize pointcloud reconstruction. As seen, purely monocular approaches perform worse than stereo-based methods, despite the median scaling aligning the predictions to the ground truth. However, incorporating the contributions from Monodepth2 (Godard et al., 2019) results in significant improvements over SfM-Learner.

Qualitative depth visualizations can be found in Figure 6. Once again, the seminal Garg et al. (2016) and SfM-Learner (Zhou et al., 2017) baselines are drastically improved w.r.t. the original implementation. However, SfM-Leaner predictions are still characterized by artifacts common to purely monocular supervision. Most notably, objects moving at similar speeds to the camera are predicted as holes of infinite depth, due to the fact that they appear static across images. Static pixel automasking from Monodepth2 (Godard et al., 2019)

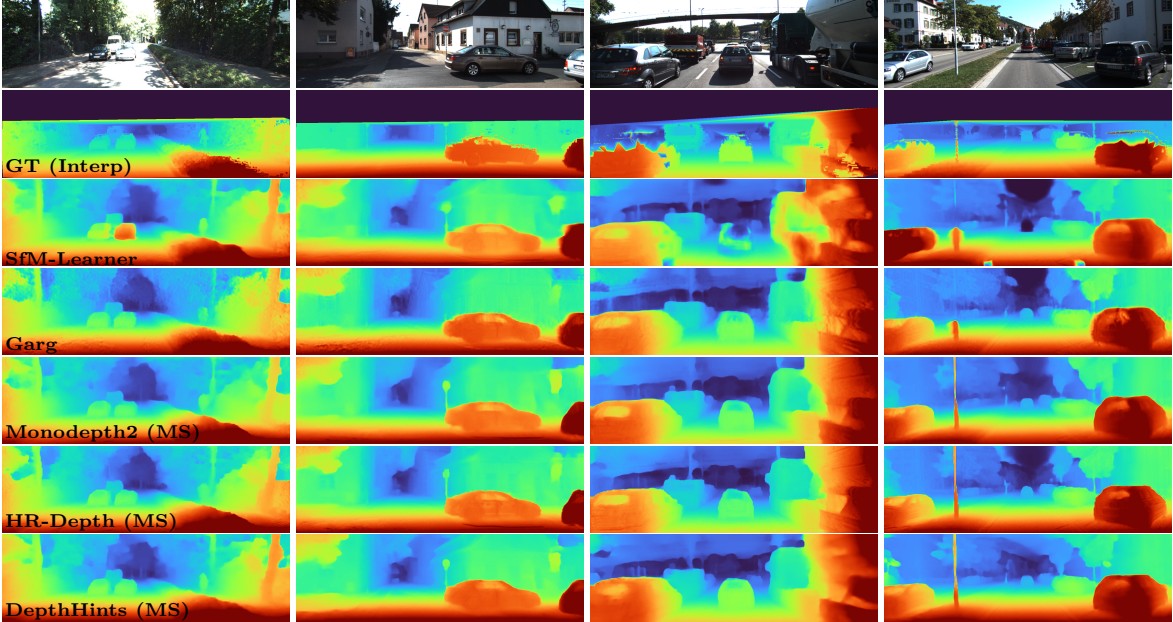

Figure 6: **Kitti Visualization.** Baseline models (Garg et al., 2016; Zhou et al., 2017) are greatly improved from their original implementations. Incorporating the minimum reconstruction loss & automasking from Monodepth2 (Godard et al., 2019) improves accuracy on thin structures and prevents holes of infinite depth. Full results in Table 6.

Table 6: **Kitti Eigen-Benchmark Evaluation.** When training in comparable conditions, the stereo baseline (Garg et al., 2016) is one of the top performing methods. The minimum reprojection and automasking losses (Godard et al., 2019) help to improve performance and mitigate monocular supervision artefacts. This is further improved via a high-resolution decoder (Lyu et al., 2021) and proxy depth supervision (Watson et al., 2019). However, these contributions only improve image-based depth metrics, but do not result in more accurate 3-D pointcloud reconstructions.

| KEB (test) | Train | Image-based | | | | | Pointcloud-based | | | |
| | | # | MAE↓ | RMSE↓ | AbsRel↓ | LogSI↓ | # | Chamfer↓ | F-Score↑ | IoU↑ |
|---|---|---|---|---|---|---|---|---|---|---|
| SfM-Learner | M | 22 | 1.98 | 4.57 | 10.69 | 15.80 | 22 | 0.73 | 44.77 | 30.03 |
| Klodt | M | 21 | 1.96 | 4.54 | 10.49 | 15.86 | 21 | 0.72 | 45.26 | 30.40 |
| Monodepth2 | M | 18 | 1.84 | 4.11 | 8.82 | 13.10 | 18 | 0.71 | 46.64 | 31.62 |
| Johnston | M | 19 | 1.83 | 3.99 | 8.85 | 12.89 | 20 | 0.71 | 45.72 | 30.78 |
| HR-Depth | M | 17 | 1.80 | 4.04 | 8.65 | 12.75 | 17 | 0.69 | 47.35 | 32.10 |
| Garg | S | _2_ | 1.60 | 3.75 | _7.65_ | 11.39 | **1** | **0.60** | **53.28** | **37.33** |
| Monodepth | S | 6 | 1.61 | 3.72 | 7.73 | 11.57 | 6 | 0.64 | 51.25 | 35.45 |
| SuperDepth | S | 9 | 1.64 | 3.77 | 7.81 | 11.63 | _2_ | 0.63 | _52.30_ | _36.40_ |
| Depth-VO-Feat | MS | 4 | 1.63 | 3.72 | 7.70 | 11.64 | 3 | 0.62 | 52.01 | 36.15 |
| Monodepth2 | MS | 10 | 1.61 | 3.62 | 7.90 | 10.99 | 7 | 0.64 | 50.50 | 34.98 |
| FeatDepth | MS | 8 | 1.60 | 3.60 | 7.80 | 11.01 | 10 | 0.65 | 49.99 | 34.51 |
| CADepth | MS | 14 | 1.63 | 3.60 | 8.09 | 10.84 | 14 | 0.66 | 49.32 | 34.06 |
| DiffNet | MS | 12 | 1.62 | 3.63 | 7.97 | 10.93 | 12 | 0.65 | 49.63 | 34.23 |
| HR-Depth | MS | 3 | _1.58_ | 3.56 | 7.70 | 10.68 | 5 | _0.62_ | 51.49 | 35.93 |
| Kuznietsov | SD* | 20 | 1.82 | 3.98 | 9.32 | 11.80 | 19 | 0.71 | 45.80 | 30.63 |
| DVSO | SD* | 13 | 1.66 | 3.77 | 8.05 | 11.32 | 11 | 0.67 | 49.82 | 34.18 |
| MonoResMatch | SD* | 16 | 1.66 | 3.75 | 8.20 | 11.31 | 16 | 0.66 | 49.01 | 33.52 |
| DepthHints | MSD* | 15 | 1.63 | 3.62 | 8.10 | 10.94 | 15 | 0.66 | 49.30 | 33.80 |
| Ours | MS | **1** | 1.59 | 3.64 | **7.63** | 11.15 | 4 | 0.62 | 51.74 | 35.97 |
| Ours (Min) | MS | 7 | 1.58 | _3.53_ | 7.79 | **10.58** | 8 | 0.63 | 50.45 | 35.01 |
| Ours (Proxy) | MSD* | 5 | **1.57** | **3.50** | 7.73 | _10.59_ | 9 | 0.64 | 50.36 | 34.69 |
| Ours (Min+Proxy) | MSD* | 11 | 1.61 | 3.56 | 7.97 | 10.74 | 13 | 0.65 | 49.46 | 33.90 |

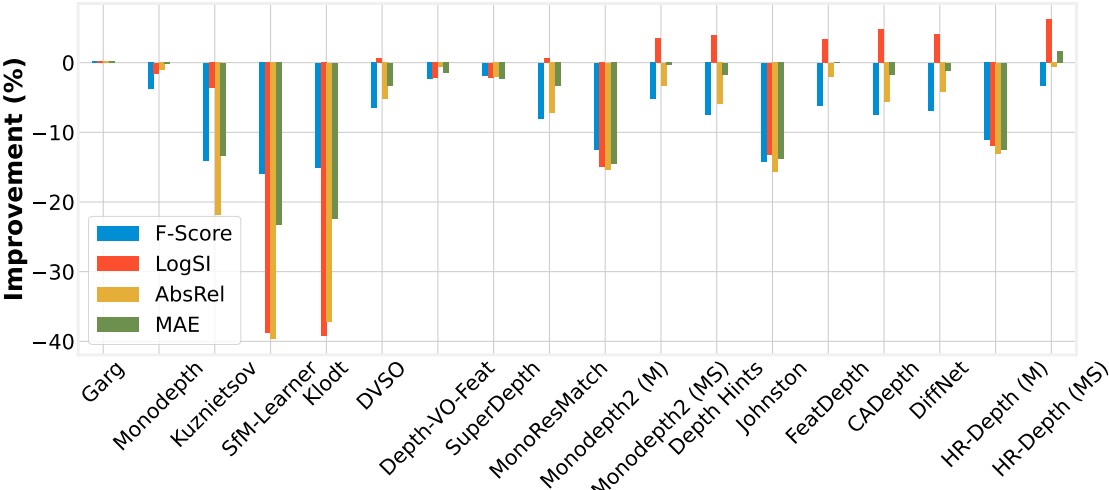

Figure 7: **Kitti Eigen-Benchmark Improvement.** When training and evaluating in fair conditions, many contributions do not result in relative improvements w.r.t. the Garg et al. (2016) stereo baseline. Most notably, all monocular-supervised approaches perform significantly worse despite the per-image median scaling. Full results in Table 6.

fixes most of these artifacts. Similarly, the minimum reconstruction loss leads to more accurate predictions for thin objects, such as traffic signs and posts. DepthHints (Watson et al., 2019) and HR-Depth (Lyu et al., 2021) independently improve the quality of the predictions via proxy depth supervision and a high-resolution decoder, respectively. However, these contributions do not significantly improve the 3-D reconstructions.

### 5.4 SYNS-Patches

Finally, we evaluate the baselines on the SYNS-Patches dataset. As discussed in Sections 3.2 & B.3, this dataset consists of 1175 images from a variety of different scenes, such as woodlands, natural scenes and urban residential/industrial areas. It is worth noting that we evaluate the models from previous section without re-training or fine-tuning. As such, SYNS-Patches represents a dataset completely unseen during training. This allows us to evaluate the robustness of the learned representations to new unseen environment types. We select a subset of 5 images per category to evaluate the qualitative performance. To make results more comparable, all models are evaluated using the monocular protocol, where each predicted depth map is aligned to the ground-truth using per-image median scaling. We reuse the metrics from the KEB split and additionally report edge-based accuracy and completion from Koch et al. (2018). Finally, we also compute the F-Score only at depth boundary pixels, reflecting the quality of the predicted discontinuities.

Results for this evaluation can be found in Table 7. As seen, performance decreases drastically for all approaches, showing that models do not transfer well beyond the automotive domain. A further decrease in F-Score can be seen when evaluating only on depth edges, indicating this as a common source of error. Once again, models incorporating recent contributions provide SotA performance in traditional image-based

Table 7: **SYNS-Patches Evaluation.** Overall performance is drastically reduced when evaluating outside the automotive training domain. Methods using minimum reconstruction loss and automasking improve image-based metrics, while Garg et al. (2016) still provides some of the top 3-D pointcloud reconstructions. Predicted edge boundaries are typically accurate (Edge-Acc <5 px), but incomplete (Edge-Comp >25 px).

| SYNS-Patches | Train | Image-Based | | | | Pointcloud-based | | | Edge-based | | | |
|---|---|---|---|---|---|---|---|---|---|---|---|---|
| | | # | MAE↓ | RMSE↓ | AbsRel↓ | # | F-Score↑ | IoU↑ | # | F-Score↑ | Acc↓ | Comp↓ |
| SfM-Learner | M | 22 | 5.43 | 9.25 | 31.58 | 22 | 11.79 | 6.43 | 20 | 8.47 | 3.46 | 36.12 |
| Klodt | M | 20 | 5.40 | 9.20 | 31.20 | 21 | 12.00 | 6.57 | 19 | 8.48 | 3.44 | 35.22 |
| Monodepth2 | M | 13 | 5.33 | 9.02 | 30.05 | 20 | 12.08 | 6.62 | 21 | 8.46 | 3.30 | 37.01 |
| Johnston | M | 10 | 5.24 | 8.92 | 29.72 | 18 | 12.16 | 6.66 | 18 | 8.60 | 3.23 | 42.82 |
| HR-Depth | M | 8 | 5.26 | 8.95 | 29.53 | 5 | 13.37 | 7.40 | 6 | 9.16 | **3.07** | 30.03 |
| Garg | S | 15 | 5.29 | 9.20 | 30.73 | 2 | 13.48 | 7.45 | **1** | **9.53** | 3.37 | 26.79 |
| Monodepth | S | 21 | 5.29 | 9.20 | 31.27 | 19 | 12.14 | 6.67 | 17 | 8.69 | 3.57 | 61.15 |
| SuperDepth | S | 16 | 5.26 | 9.08 | 30.83 | 13 | 12.87 | 7.10 | 11 | 9.01 | 3.40 | 40.40 |
| Depth-VO-Feat | MS | 17 | 5.30 | 9.17 | 30.83 | 16 | 12.43 | 6.82 | 15 | 8.77 | 3.50 | 38.49 |
| Monodepth2 | MS | 6 | 5.18 | 8.91 | 29.04 | 8 | 13.18 | 7.27 | 13 | 8.95 | 3.38 | 32.69 |
| FeatDepth | MS | 7 | 5.16 | 8.80 | 29.12 | 17 | 12.27 | 6.73 | 22 | 8.41 | 3.50 | 44.09 |
| CADepth | MS | 11 | 5.22 | 8.97 | 29.80 | 14 | 12.83 | 7.06 | 16 | 8.70 | 3.42 | 35.89 |
| DiffNet | MS | 2 | 5.16 | 8.91 | 28.80 | 9 | 13.16 | 7.26 | 14 | 8.81 | 3.45 | 39.46 |
| HR-Depth | MS | 5 | 5.13 | 8.85 | 28.94 | **1** | **13.79** | **7.65** | 5 | 9.21 | 3.25 | 28.33 |
| Kuznietsov | SD* | 19 | 5.47 | 9.50 | 31.08 | 10 | 13.15 | 7.26 | 7 | 9.11 | 3.39 | 47.13 |
| DVSO | SD* | 9 | 5.18 | 8.93 | 29.66 | 11 | 13.08 | 7.23 | 2 | 9.29 | 3.34 | 40.23 |
| MonoResMatch | SD* | 14 | 5.24 | 9.07 | 30.28 | 15 | 12.73 | 7.01 | 9 | 9.03 | 3.47 | 51.03 |
| DepthHints | MSD* | 18 | 5.33 | 9.07 | 30.90 | 12 | 12.91 | 7.11 | 10 | 9.01 | 3.24 | **26.21** |
| Ours | MS | 12 | 5.20 | 9.03 | 29.93 | 4 | 13.38 | 7.39 | 3 | 9.28 | 3.31 | 34.36 |
| Ours (Min) | MS | **1** | 5.11 | 8.80 | **28.59** | 7 | 13.20 | 7.27 | 12 | 8.98 | 3.24 | 32.46 |
| Ours (Proxy) | MSD* | 3 | 5.11 | 8.79 | 28.87 | 3 | 13.46 | 7.45 | 4 | 9.23 | 3.16 | 30.60 |
| Ours (Min+Proxy) | MSD* | 4 | **5.08** | **8.71** | 28.91 | 6 | 13.23 | 7.30 | 8 | 9.11 | 3.16 | 31.32 |

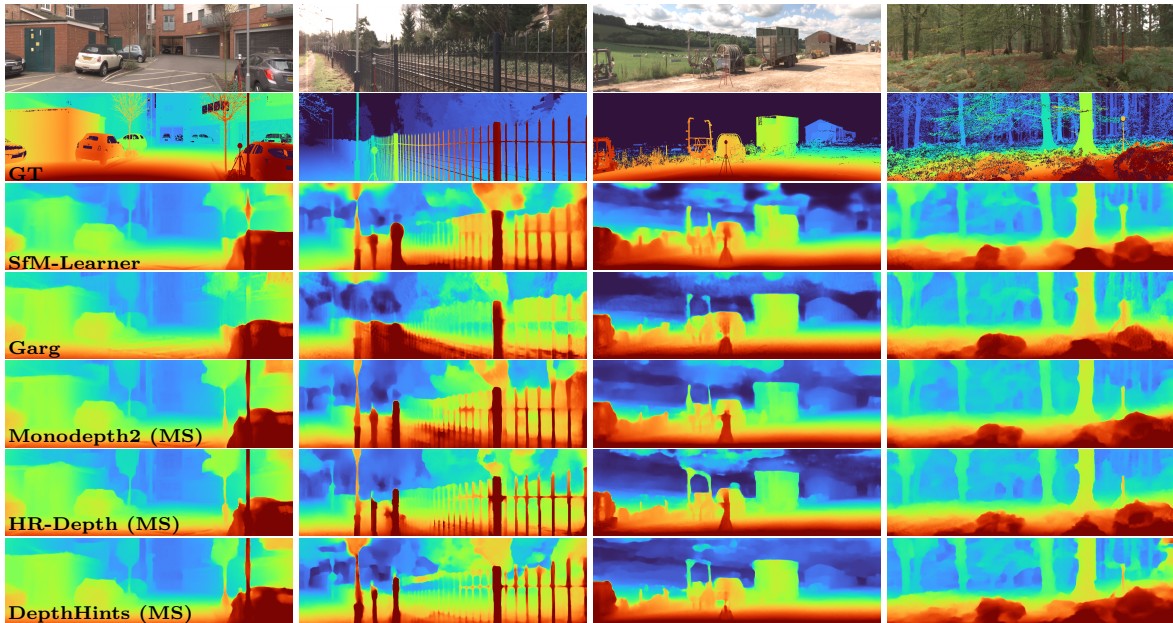

Figure 8: **SYNS Visualization.** As evidenced by Table 7, models trained on Kitti do not transfer well to natural scenes, such as woodlands. Similar to Kitti, we find the contributions from Mondepth2 (Godard et al., 2019) to improve prediction accuracy in challenging areas such as thin structures and object boundaries. Full results in Table 7.

depth metrics. However, Garg et al. (2016) consistently remains one of the top performers in 3-D pointcloud reconstruction and edge-based metrics. In general, predicted edges are typically accurate (∼3 px error). However, there are many missing edges, as reflected by the large edge completeness error (∼26 px error).

Qualitative depth visualizations can be found in Figure 8. Similar to Kitti, Monodepth2 (Godard et al., 2019) and its successors (Watson et al., 2019; Lyu et al., 2021) greatly improve performance on thin structures. However, there is still room for improvement, as shown by the railing prediction in the second image. This is reflected by the low 3-D reconstruction metrics. Furthermore, all methods perform significantly worse in natural and woodlands scenes, demonstrating the need for more varied training data.

## 6 Conclusion

This paper has presented a detailed ablation procedure to critically evaluate the current SotA in self-supervised monocular depth learning. We independently reproduced 16 recent baselines, modernizing them with sensible design choices that improve their overall performance. When benchmarking on a level playing field, we show how many contributions do not provide improvements over the legacy stereo baseline. Even in the case where they do, the change in performance is drastically lower than that claimed by the original publications. This results in new SotA models that set the bar for future contributions. Furthermore, this work has shown how, in many cases, the silent changes that are rarely claimed as contributions can result in equal or greater improvements than those provided by the claimed contribution.

Regarding future work, we identify two main research directions. The first of these is the generalization capabilities of monocular depth estimation. Given the results on SYNS-Patches, it is obvious that purely automotive data is not sufficient to generalize to complex natural environments, or even other urban environments. As such, it would be of interest to explore additional sources of data, such as indoor sequences, natural scenes or even synthetic data. The second avenue should focus on the accuracy on thin objects and depth discontinuities, which are challenging for all existing methods. This is reflected in the low F-Score and Edge Completion metrics in these regions. To aid future research and encourage good practices, we make the proposed codebase publicly available. We invite authors to contribute to it and use it as a platform to train and evaluate their contributions.

**Acknowledgments**

This work was partially funded by the EPSRC under grant agreements EP/S016317/1 & EP/S035761/1.

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

# A    Monocular Depth Overview

We provide a review of self-supervised monocular depth estimation, which serves as an introduction to the field while providing in-depth details about the implemented baseline models. We also provide some practical implementation details that are frequently omitted. For the purpose of this paper, we limit our benchmark to methods that perform only depth estimation along with optional VO prediction. We do not include methods that additionally learn other tasks, such as semantic segmentation, optical flow or surface normals.

## A.1    DepthNet

The core component of the framework is the depth prediction network, composed of a fully-convolutional encoder-decoder architecture with skip connections. This network produces multi-scale dense depth predictions.

**Encoder.** Any architecture producing a multi-scale feature representation can be used as the encoder. In practice, this is implemented in a highly flexible way by using the `timm` library (Wightman, 2019), containing pretrained models for recent SotA classification architectures. As part of the ablation presented in Section 4 we perform an in-depth study of the most effective backbone architecture and its pretraining method. We find ConvNeXt (Liu et al., 2022) to provide the best performance overall.

**Depth Decoder.** The commonly used dense decoder is inspired by the DispNet architecture (Mayer et al., 2016). It contains five upsampling stages, each composed of two Conv-ELU blocks with nearest-neighbour upsampling and a concatenated skip connection from the corresponding encoder stage. Each stage makes an initial depth prediction at its reduced resolution, used as additional supervision during training. These intermediate predictions are omitted from the following network and loss equations for the sake of brevity and clarity. In practice, DepthNet predicts the inverse depth (*i.e.* disparity) due to its increased stability. As such, this prediction is obtained via a convolution followed by a sigmoid activation.

The depth prediction process can be summarized as

$$\hat{\mathbf{Z}} = \Phi_D\left(\mathbf{I}\right), \tag{1}$$

where $\Phi_D$ is the full DepthNet architecture and $\hat{\mathbf{Z}}$ is the predicted sigmoid disparity. This is converted into a scaled depth prediction through

$$\hat{\mathbf{D}} = \frac{1}{a\hat{\mathbf{Z}} + b}, \tag{2}$$

where $a$ & $b$ are constants such that the final depth is in the range $[0.1, 100]$. It is worth noting this scale is arbitrary and does not correspond to metric depth.

**Virtual Stereo.** Monodepth (Godard et al., 2017) introduced the concept of a virtual stereo prediction, where the network is forced to predict both the left and right disparity from only the left input image. This is achieved by adding an extra output channel to the disparity prediction at each encoder scale. The original Monodepth was trained using only images from the left viewpoint, meaning the virtual disparity would always correspond to the right view.

To make this procedure more flexible and allow for training with either stereo viewpoint, we adopt a procedure similar to 3Net (Poggi et al., 2018). We assume the input to the network is the central viewpoint in a three-camera rig and predict two virtual disparities, corresponding to the left/right cameras. During training, we select the "real virtual" disparity, *i.e.* if the network was given the right image we sample the virtual left disparity, and vice versa. It is worth noting this does not require an additional decoder as in Poggi et al. (2018), since we simply add two extra output channels to each network prediction.

**Predictive Mask.** Training in a purely monocular setting (Zhou et al., 2017; Klodt & Vedaldi, 2018) is susceptible to artifacts not present during stereo training. Dynamic objects moving independently from the rest of the scene are unaccounted for in the predicted PoseNet motion, resulting in incorrect synthesized views even if the depth prediction is correct. If the object is moving at similar speeds to the camera, it will appear as static between images, resulting in predictions of infinite depth. Meanwhile, objects moving towards the camera will seem closer than they are and depth will be underestimated.

To mitigate these effects, SfM-Learner (Zhou et al., 2017) introduced an additional decoder to predict an explainability mask in the range $[0, 1]$. This mask was trained to downweight the photometric loss in unreliable regions, such as dynamic objects or specularities and regularized using a binary cross-entropy loss pushing all mask values towards 1. Klodt & Vedaldi (2018) instead adopted the uncertainty formulation introduced by Kendall & Gal (2017). The network is therefore trained to predict the log variance uncertainty associated with each photometric error pixel. In this case, the mask is restricted to positive uncertainty values.

Following recent implementations (Godard et al., 2019), these masks are predicted for each of the support images via an additional DepthNet decoder. Details on integrating these masks into the photometric loss are provided in Section A.4.

## A.2 PoseNet

**Encoder.** Similar to DepthNet, the encoder can be easily replaced with any desired architecture by leveraging the pretrained models from Wightman (2019). However, PoseNet is typically restricted to a pretrained ResNet-18 for efficiency. VO requires us to predict the *relative* motion between frames. As such, we perform an independent prediction for each support frame by concatenating them channel-wise with the target frame. We modify the first convolutional layer of the architecture accordingly, ensuring the pretrained weights are correctly duplicated and scaled. As is common, we force the network to make a forward-motion prediction by swapping the image order if necessary.

**Decoder.** Since VO is a holistic regression task (it produces one output for the whole image), we use a simple decoder. The encoder features are projected to a lower dimensionality via a $1 \times 1$ convolution and processed by two Conv-ReLU blocks. The final prediction is obtained by applying a convolution without an activation. As is common practice, we scale the network predictions by a factor of 0.01 to improve stability and convergence. The network produces a 6-D output, formed by a translation and axis-angle rotation. The rotation vector is converted into a $3 \times 3$ transform matrix via the Rodrigues formula, with the magnitude indicating the angle and the direction corresponding to the axis of rotation.

Overall, the process for obtaining the predicted motion $\hat{\mathbf{P}}_{t \to t+k}$ between frames $\mathbf{I}_t$ & $\mathbf{I}_{t+k}$ is simplified as

$$\hat{\mathbf{P}}_{t \to t+k} = \Phi_P \left( \mathbf{I}_t \oplus \mathbf{I}_{t+k} \right), \tag{3}$$

where $\oplus$ represents channel-wise concatenation and $\Phi_P$ is the resulting PoseNet.

## A.3 View Synthesis

The core component of all self-supervised monocular depth approaches is the ability to synthesize the target image from a set of adjacent support frames. Given the scene depth and camera locations, a point in the target image $\mathbf{p}_t$ can be reprojected onto any of the available frames as

$$\mathbf{p}'_{t+k} = \mathbf{K}\hat{\mathbf{P}}_{t \to t+k}\hat{\mathbf{D}}_t \left( \mathbf{p}_t \right) \mathbf{K}^{-1}\mathbf{p}_t, \tag{4}$$

where $\mathbf{K}$ represents the shared camera intrinsic parameters, $\hat{\mathbf{D}}_t \left( \mathbf{p}_t \right)$ is the predicted depth at the given point and $\hat{\mathbf{P}}_{t \to t+k}$ is the predicted camera motion matrix (or the known stereo baseline). Using these correspondences, it is possible to synthesize a support frame aligned to the target image by bilinearly sampling at each image pixel. This is given by

$$\mathbf{I}'_{t+k} = \mathbf{I}_{t+k} \left\langle \mathbf{p}'_{t+k} \right\rangle, \tag{5}$$

where $\langle \cdot \rangle$ represents bilinear sampling using Spatial Transformer Networks (Jaderberg et al., 2015).

## A.4 Losses

**Photometric.** The main loss is the photometric error. This measures the difference between the raw target frame $\mathbf{I}_t$ and the synthesized view from the support frame $\mathbf{I}'_{t+k}$ generated by (5). In the most simple case,

an $L_1$ loss can be used. However, this is typically complemented by an SSIM (Shi et al., 2016) loss to improve its robustness. This is defined as

$$\mathcal{L}_{photo}\left(\mathbf{I}, \mathbf{I}'\right) = \lambda_{ssim}\frac{1\text{-}\mathcal{L}_{ssim}\left(\mathbf{I}, \mathbf{I}'\right)}{2} + (1\text{-}\lambda_{ssim})\,\mathcal{L}_1\left(\mathbf{I}, \mathbf{I}'\right), \tag{6}$$

where $\lambda_{ssim}$ represents the SSIM weight. This photometric loss can be additionally downweighted when using a predictive mask. When using the explainability mask $\mathbb{M}_E$ (Zhou et al., 2017) this is simply defined as

$$\mathcal{L}_{photo}\left(\mathbf{I}, \mathbf{I}'\right) = \mathbb{M}_E \odot \mathcal{L}_{photo}\left(\mathbf{I}, \mathbf{I}'\right), \tag{7}$$

while the uncertainty mask $\mathbb{M}_U$ (Klodt & Vedaldi, 2018) affects it as

$$\mathcal{L}_{photo}\left(\mathbf{I}, \mathbf{I}'\right) = \exp\left(\text{-}\mathbb{M}_U\right) \odot \mathcal{L}_{photo}\left(\mathbf{I}, \mathbf{I}'\right) + \mathbb{M}_U, \tag{8}$$

where $\odot$ represents the Hadamard product.

**Reconstruction.** The photometric loss is aggregated by the reconstruction loss. The base case simply averages the photometric loss over each support frame and each image pixel, defined as

$$\mathcal{L}_{rec}\left(\mathbf{I}_t\right) = \overline{\sum_k} \overline{\sum_{\mathbf{p}}} \mathcal{L}_{photo}\left(\mathbf{I}_t, \mathbf{I}'_{t+k}\right), \tag{9}$$

where $\overline{\sum}_{i=0}^N i \equiv \frac{1}{N}\sum_{i=0}^N i$ represents the average summation. Monodepth2 (Godard et al., 2019) showed the benefit of instead selecting the minimum loss per-pixel over the various support frames. Intuitively, this selects correct pixel-wise correspondences, instead of averaging out incorrect errors caused by occlusions. This is simplified as

$$\mathcal{L}_{rec}\left(\mathbf{I}_t\right) = \overline{\sum_{\mathbf{p}}} \min_k \mathcal{L}_{photo}\left(\mathbf{I}_t, \mathbf{I}'_{t+k}\right). \tag{10}$$

Monodepth2 additionally proposed an automatic masking procedure. Static pixels are removed from the loss if the photometric error for the original support frame is lower than the synthesised view loss. This is given by

$$\mathbb{M}_S = \left[\!\!\left[ \min_k \mathcal{L}_{photo}\left(\mathbf{I}_t, \mathbf{I}'_{t+k}\right) < \min_k \mathcal{L}_{photo}\left(\mathbf{I}_t, \mathbf{I}_{t+k}\right) \right]\!\!\right], \tag{11}$$

where $[\!\![\cdot]\!\!]$ represents the Iverson brackets and $\mathbb{M}_S$ is the resulting mask selecting non-static pixels. Note that the second term in the equation uses the original support frame $\mathbf{I}_{t+k}$.

**Feature reconstruction.** As noted by previous works (Zhan et al., 2018; Spencer et al., 2020; Shu et al., 2020), the image-based reconstruction loss might still produce ambiguous correspondences. This is especially the case in low-light environments that contain multiple light sources and complex reflections. In this case, it can be beneficial to incorporate an additional feature-based reconstruction loss, defined as

$$\mathcal{L}_{feat}\left(\mathbf{F}_t\right) = \overline{\sum_{\mathbf{p}}} \min_k \mathcal{L}_{photo}\left(\mathbf{F}_t, \mathbf{F}'_{t+k}\right), \tag{12}$$

$$\mathbf{F}'_{t+k} = \mathbf{F}_{t+k}\left\langle \mathbf{p}'_{t+k}\right\rangle, \tag{13}$$

where $\mathbf{F}'_{t+k}$ are the synthesized support features using the previously computed reprojection correspondences from (4). Note that in this case the $L_1$ + SSIM photometric loss can be replaced with the $L_2$ distance between feature embeddings. In practice, instead of introducing an additional pretrained dense feature network, we propose to re-use the low-level features from the depth encoder.

**Depth regression.** To complement the self-supervised reconstruction losses, previous approaches have introduced an additional proxy supervised regression loss. This can prevent local minima during the optimization process. When computing the loss between disparities we opt for the standard $L_1$ loss. However, when in

depth space we use other well-established losses. The first of these is the reverse Huber (berHu) loss (Zwald & Lambert-Lacroix, 2012; Laina et al., 2016), defined as

$$\mathcal{L}_{berHu}\left(\hat{\mathbf{D}},\mathbf{D}\right) = \begin{cases} \mathcal{L}_{1}\left(\hat{\mathbf{D}},\mathbf{D}\right) & \text{where } \mathcal{L}_{1}\left(\hat{\mathbf{D}},\mathbf{D}\right) \leq \delta \\ \dfrac{\mathcal{L}_{1}\left(\hat{\mathbf{D}},\mathbf{D}\right)+\delta^2}{2\delta} & \text{otherwise} \end{cases}, \tag{14}$$

where $\mathbf{D}$ represents the (proxy) ground truth depth, $\hat{\mathbf{D}}$ is the network prediction and the margin threshold is adaptively set per-batch as $\delta = 0.2\max\mathcal{L}_{1}\left(\hat{\mathbf{D}},\mathbf{D}\right)$. DepthHints (Watson et al., 2019) argues the berHu loss is better suited towards regressing ground-truth LiDAR and proposes to use the Log-L$_1$ loss when regressing hand-crafted stereo disparities, given by

$$\mathcal{L}_{logL_1}\left(\hat{\mathbf{D}},\mathbf{D}\right) = \log\left(1 + \mathcal{L}_{1}\left(\hat{\mathbf{D}},\mathbf{D}\right)\right). \tag{15}$$

**Virtual stereo consistency.** Following Monodepth (Godard et al., 2017), we use the virtual stereo consistency loss, forcing the network to predict a consistent scene depth for both viewpoints. This is achieved by applying the view synthesis module to the *disparities*, as opposed to the images. This results in

$$\mathcal{L}_{stereo}\left(\hat{\mathbf{Z}}_t,\hat{\mathbf{Z}}'_{t+k}\right) = \mathcal{L}_{1}\left(\hat{\mathbf{Z}},\hat{\mathbf{Z}}'_{t+k}\right), \tag{16}$$

$$\hat{\mathbf{Z}}'_{t+k} = \hat{\mathbf{Z}}_{t+k}\left\langle\mathbf{p}'_{t+k}\right\rangle, \tag{17}$$

where $\hat{\mathbf{Z}}'_{t+k}$ is the warped disparity corresponding to the virtual stereo prediction from the network.

## A.5 Regularization

**Disparity smoothness.** In general, we expect a pixel's disparity to be similar to that of its neighbours. Garg et al. (2016) introduced a simple regularization constraint encouraging this by penalizing all gradients in the disparity map, defined as

$$\mathcal{L}_{smooth}\left(\hat{\bar{\mathbf{Z}}}\right) = \sum_{\mathbf{p}}\left|\partial\hat{\bar{\mathbf{Z}}}\left(\mathbf{p}\right)\right|, \tag{18}$$

where $\partial\hat{\bar{\mathbf{Z}}}$ are the spatial gradients of the mean-normalized disparity, computed in the $x$ & $y$ directions independently. We omit these directions from the equations for clarity. Godard et al. (2017) softened this constraint by allowing disparity gradients proportional to the image gradients at that pixel location. This is known as the edge-aware smoothness regularization, given by

$$\mathcal{L}_{smooth}\left(\hat{\bar{\mathbf{Z}}}\right) = \sum_{\mathbf{p}}\left|\partial\hat{\bar{\mathbf{Z}}}\left(\mathbf{p}\right)\right|\exp\left(-\left|\partial\mathbf{I}\left(\mathbf{p}\right)\right|\right). \tag{19}$$

Rui et al. (2018) instead enforce a smooth change in gradients, *i.e.* second-order smoothness. This is defined as

$$\mathcal{L}_{smooth}\left(\hat{\bar{\mathbf{Z}}}\right) = \sum_{\mathbf{p}}\left|\partial^2\hat{\bar{\mathbf{Z}}}\left(\mathbf{p}\right)\right|\exp\left(-\left|\partial^2\mathbf{I}\left(\mathbf{p}\right)\right|\right), \tag{20}$$

where $\partial^2\hat{\bar{\mathbf{Z}}}$ represents the second-order gradients in each spatial direction. In this benchmark we add the option to apply Gaussian smoothing prior to computing the disparity/image gradients.

**Occlusions.** DVSO (Rui et al., 2018) noted that the smoothness constraints can result in oversmoothed predictions. To counteract this, they proposed an occlusion regularization term that penalizes the sum of disparities in the scene. This should allow the network to predict sharper boundaries at occlusions, while favouring background disparities. This is simply defined as

$$\mathcal{L}_{occ}\left(\hat{\mathbf{Z}}\right) = \sum_{\mathbf{p}}\hat{\mathbf{Z}}\left(\mathbf{p}\right). \tag{21}$$

**Masks.** Using the explainability mask (Zhou et al., 2017) lets the network downweigh the photometric loss in regions where it believes correspondences may be incorrect or uninformative. This occurs with dynamic object moving independently from the scene. To avoid the degenerate case where the mask ignores all pixels, an additional binary cross-entropy regularization pushes all values towards 1.

## B   Benchmark Datasets

Complementing Section 3, we provide additional details for each of dataset. This includes the various metrics defined by the Kitti splits, as well as the creation procedure for the SYNS-Patches dataset.

### B.1   Kitti Eigen

As discussed in the main paper, we strongly encourage future authors to avoid this evaluation, due to the inaccurate ground-truth. However, we provide details for comparison with previous publications as the community transitions to the proposed benchmark. The KE split (Eigen & Fergus, 2015) defines the following metrics:

**AbsRel.** Measures the mean relative error (%) as

$$e = \sum \frac{|\hat{y} - y|}{y}, \tag{22}$$

where $\sum_{i=0}^{N} i \equiv \frac{1}{N} \sum_{i=0}^{N} i$ represents the average summation.

**SqRel.** Measures the mean relative square error (%) as

$$e = \sum \frac{|\hat{y} - y|^2}{y}. \tag{23}$$

Note that this is incorrectly computed, as the squared term is missing from the denominator.

**RMSE.** Measures the root mean square error (meters) as

$$e = \sqrt{\sum |\hat{y} - y|^2}. \tag{24}$$

**LogRMSE.** Measures the root mean square log error (log-meters) as

$$e = \sqrt{\sum |\log(\hat{y}) - \log(y)|^2}. \tag{25}$$

**Threshold accuracy.** Measures the threshold accuracy (%) $\delta < 1.25^\kappa$, where $\kappa \in \{1, 2, 3\}$ and

$$\delta = \max\left(\frac{\hat{y}}{y}, \frac{y}{\hat{y}}\right). \tag{26}$$

### B.2   Kitti Eigen-Benchmark

As a replacement for KE, we propose to use the updated and corrected ground-truth from Uhrig et al. (2018). This benchmark defines the following metrics:

**MAE.** Measures the mean absolute error (meters) as

$$e = \sum |\hat{y} - y|. \tag{27}$$

**RMSE.** Measures the root mean square error (meters) as

$$e = \sqrt{\sum |\hat{y} - y|^2}. \tag{28}$$

**InvMAE.** Measures the mean absolute inverse error (1/meters) as

$$e = \sum \left| \frac{1}{\hat{y}} - \frac{1}{y} \right|. \tag{29}$$

**InvRMSE.** Measures the root mean square inverse error (1/meters) as

$$e = \sqrt{\sum \left| \frac{1}{\hat{y}} - \frac{1}{y} \right|^2}. \tag{30}$$

**LogMAE.** Measures the mean absolute log error (log-meters) as

$$e = \sum \left| \log(\hat{y}) - \log(y) \right|. \tag{31}$$

**LogRMSE.** Measures the root mean square log error (log-meters) as

$$e = \sqrt{\sum \left| \log(\hat{y}) - \log(y) \right|^2}. \tag{32}$$

**LogSI.** Measures the root mean scale invariant log error (Eigen & Fergus, 2015) (log meters) as

$$e = \sqrt{\sum \left| \log(\hat{y}) - \log(y) \right|^2 - \left( \sum \log(\hat{y}) - \log(y) \right)^2}. \tag{33}$$

Note that the second term in this metric is directional, *i.e.* the prediction is rewarded if it is consistently incorrect in the same direction. This accounts for the case where the prediction is correct, but scaled differently to the ground truth.

**AbsRel.** Measures the mean relative error (%) as

$$e = \sum \frac{|\hat{y} - y|}{y}. \tag{34}$$

**SqRel.** Measures the mean relative square error (%) as

$$e = \sum \frac{|\hat{y} - y|^2}{y^2}. \tag{35}$$

The metrics described above measure the error in the predicted 2-D depth map. However, the true objective of monocular depth estimation is to accurately reconstruct the 3-D world. As such, Örnek et al. (2022) proposed to instead evaluate depth prediction using well-established 3-D metrics, which measure the fidelity of the reconstructed pointcloud. We report the following 3-D metrics:

**Chamfer.** Measures the Chamfer distance between the reconstructed pointclouds (m) as

$$e = \sum_{\mathbf{q} \in Q} \min_{\hat{\mathbf{q}} \in \hat{Q}} \|\mathbf{q} - \hat{\mathbf{q}}\| + \sum_{\hat{\mathbf{q}} \in \hat{Q}} \min_{\mathbf{q} \in Q} \|\mathbf{q} - \hat{\mathbf{q}}\|, \tag{36}$$

where $Q$ & $\hat{Q}$ represent the ground-truth and predicted pointclouds and $\mathbf{q}$ & $\hat{\mathbf{q}}$ are 3-D points in those pointclouds, respectively.

**Precision.** Measures the number of predicted points (%) within a distance threshold $\delta$ to the ground-truth surface as

$$P = \sum_{\hat{\mathbf{q}} \in \hat{Q}} \left[\!\left[ \min_{\mathbf{q} \in Q} \|\mathbf{q} - \hat{\mathbf{q}}\| < \delta \right]\!\right]. \tag{37}$$

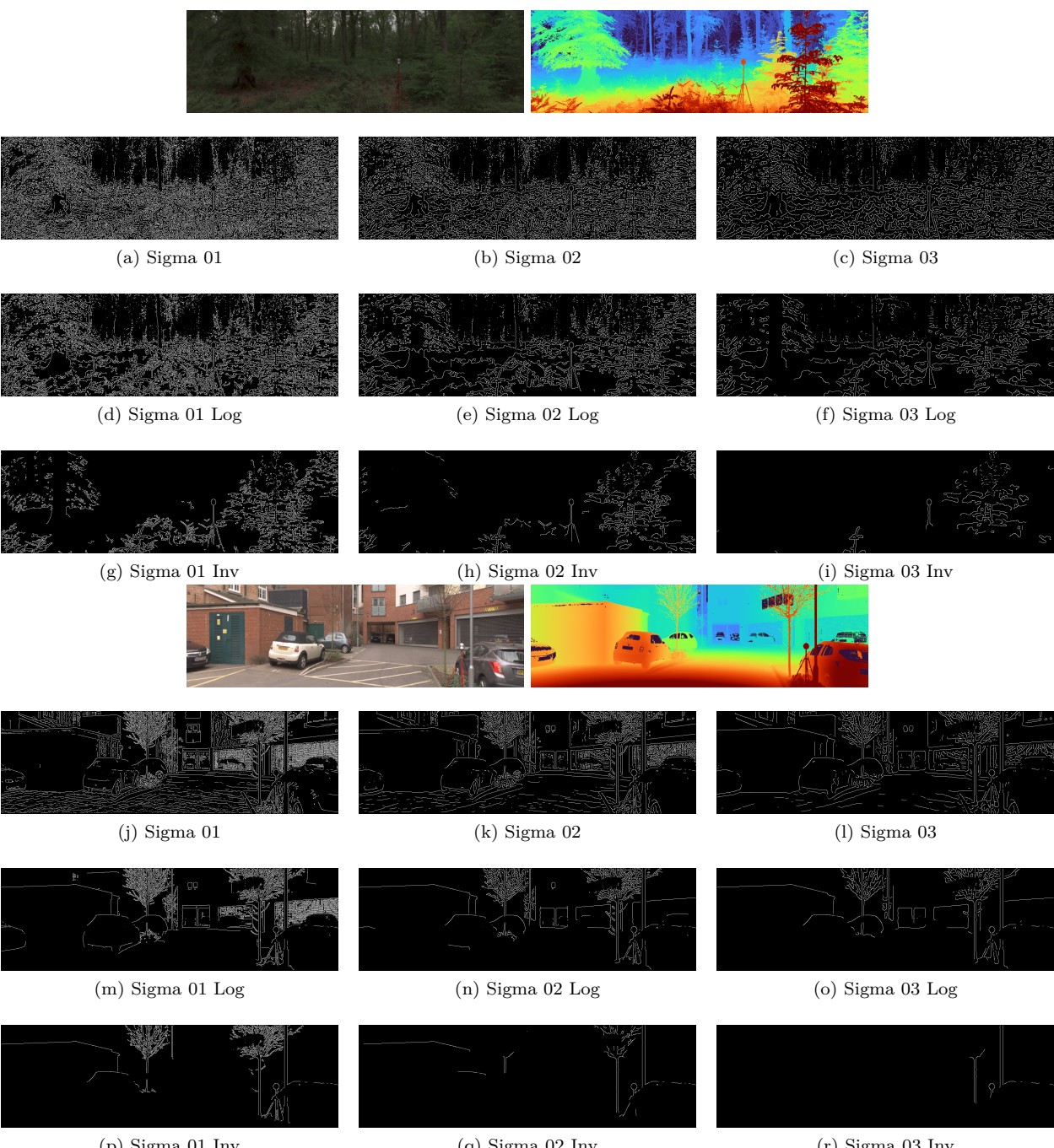

Figure 9: **SYNS-Patches Dataset.** Extracted ground-truth depth boundaries on raw/log/inv depth maps using increasing Gaussian blurring sigmas. Sigma 01 log-depth provides a good balance of edges-to-noise ratio.

**Recall.** Measures the number of ground-truth points (%) within a distance threshold $\delta$ to the predicted surface as

$$R = \sum_{\mathbf{q} \in Q} \left[\!\!\left[ \min_{\hat{\mathbf{q}} \in \hat{Q}} \| \mathbf{q} - \hat{\mathbf{q}} \| < \delta \right]\!\!\right]. \tag{38}$$

As per Örnek et al. (2022), we set the threshold for a correctly reconstructed point to $\delta = 0.1$, *i.e.* 10 cm.

**F-Score.** Also known as the Dice coefficient. Measures the harmonic mean of precision and recall (%) as

$$F = 2 \cdot \frac{P \cdot R}{P + R}. \tag{39}$$

**IoU.** Also known as the Jaccard Index between two pointclouds. Measures the volumetric quality of a 3-D reconstruction (%) as

$$IoU = \frac{P \cdot R}{P + R - P \cdot R}. \tag{40}$$

## B.3 SYNS-Patches

We generate SYNS-Patches, based on SYNS (Adams et al., 2016), by sampling undistorted patches from the spherical panorama image every 20 azimuth degrees at a constant elevation of zero. In other words, we perform a full horizontal rotation roughly at eye level, sampling 18 images per scene.

The original panoramic data was collected using a scanning camera rotating around its vertical axis, with each scan taking approximately 5-10 minutes. The LiDAR was captured after the HDR panorama following a similar procedure, resulting in dense depth scans. Since the panorama is not instantly captured, it is susceptible to distortions from moving objects. Furthermore, since the image and depth scans were captured independently, it is possible to have mismatched "static-but-dynamic" objects, such as parked cars or stationary pedestrians. We visually check the extracted dataset and remove images containing these artifacts, along with empty/uniform depth maps. The final test set contains 1,175 out of the possible $18 \times 92 = 1,656$ images.

Ground-truth depth boundaries are obtained by detecting Canny edges in the depth map. This is sensitive to missing data (due to infinite depth or highly reflective surfaces), which can cause fake boundaries. We take this into account by masking edges connected to regions with invalid depths, unless these pixels belong to the sky. Accurate sky masks are obtained using a pretrained SotA semantic segmentation model (Reda et al., 2018). We generate depth edges at three levels of Gaussian smoothing, using either raw, log or inverse depth maps. In practice, we find log-depth and a smoothing sigma of 1 to provide the best qualitative results. Example depth boundary predictions for different hyperparameters are shown in Figure 9.

As in KEB, we use the original image-based metrics defined by Uhrig et al. (2018), complemented by the pointcloud-based metrics from Örnek et al. (2022). To provide more granular results, we optionally compute all metrics only at the detected ground-truth depth boundaries. We additionally report the edge-based metrics of Koch et al. (2018):

**EdgeAcc.** Measures the accuracy of the predicted depth edges (pixels) via the distance from each predicted edge to the closest ground-truth edge as

$$e = \sum_P EDT\left(\hat{\mathbf{Y}}_{bin}\left(\mathbf{p}\right)\right) : P = \{\mathbf{p} | \mathbf{Y}_{bin}\left(\mathbf{p}\right) = 1\}, \tag{41}$$

where $EDT$ represents the Euclidean Distance Transform, truncated to a maximum distance $\delta = 10$ and $\mathbf{Y}_{bin}$ & $\hat{\mathbf{Y}}_{bin}$ are the binary maps of depth boundaries for the ground-truth and prediction, respectively.

**EdgeComp.** Measures the completeness of the predicted depth edges (pixels) via the distance from each ground-truth edge to the closest predicted edge as

$$e = \sum_P EDT(\mathbf{Y}_{bin}\left(\mathbf{p}\right)) : P = \left\{\mathbf{p} | \hat{\mathbf{Y}}_{bin}\left(\mathbf{p}\right) = 1\right\}. \tag{42}$$

