# OpenReview forum: "Deconstructing Self-Supervised Monocular Reconstruction: The Design Decisions that Matter"
_TMLR — Accepted by TMLR_

### Review · Reviewer_YWtU · 2022-09-20

**Summary Of Contributions:**

This submission analyzes recent progress on self-supervised monocular depth estimation task. It aims to decouple the improvements brought by better architecture and better loss functions. To do so, it re-implements 16 recent methods within the same codebase, and performs systematic study on network architecture and loss functions. The experiments are performed on a newly-introduced depth estimation benchmark, which includes the error-corrected KITTI dataset and a new dataset called SYNS-patches (containing 1k+ images of various urban and natural scenes with high-quality ground-truth). One surprising finding is that simply replacing the backbone with an up-to-date model results in a much bigger gain, compared to the improvements brought by loss functions.

**Broader Impact Concerns:**

No.

**Requested Changes:**

**Besides the points in the weakness**
- P14 conclusion “Even in the case where they do, the change is performance is drastically” => "the change of performance is..."
- Tab. 6 is out of the page margin. May consider resizing the table or reducing the number of columns.
- Intro-paragraph “ It is a core component enabling mid-level tasks such as Structure-from-Motion” -- does depth estimation help sfm, which by its definition takes correspondences as input instead of depth?

**Changes that would strengthen the work**
- In Fig 4, besides accuracy, another angle to look at the backbone ablation is the speed / training iterations.
- Would be helpful to provide more details on the SYNS-patches dataset in terms of depth range (of the ground-truth) and density. Which depth metric is sensitive to depth range, and which are range-invariant?

- Arguably, if calibrated stereo inputs are used at training time, it is not fair to call them self-supervised depth prediction, since ground-truth depth can be easily computed by triangulation / stereo matching algorithm. In that sense, another baseline for all the self-supervised methods would be “converting” stereo input to depth “pseudo”-ground truth, and then apply an additional supervised losses.


**Strengths And Weaknesses:**

**Strengths**
- A benchmark with standardized/correct depth metrics is introduced. The proposed SYNS-Patches dataset seems to have high-quality depth ground-truth with fine details.
- The ablation study is thorough, which includes backbone architecture/pre-training and depth regularization losses. The finding made by the paper that recent improvements in self-supervised depth estimation methods comes mostly from a better backbone network is interesting. This suggests research in this field needs to consider isolating the effect of backbone network when claiming for contributions in self-supervised losses.

**Weaknesses**
- Given the large number of methods that the submission aims to analyze, the paper didn't do a good job organizing and differentiating them. In its current state, it is difficult to keep track of the contributions each method made. A table of related work would help summarizing/grouping/differentiating their contributions in this case. For instance, what type of loss each method proposed / used and which backbone each method applied. The losses proposed by each method are sparsely discussed in Sec 4.3/Sec 5.1, and can be merged into a single table.
- The task of self-supervised depth estimation is not well-motivated. For instance, given there are high-quality depth ground-truth (e.g., Omnidata/MiDAS), and supervised depth estimation methods available, why would one care about self-supervised depth estimation task.
- Training dataset is limited to KITTI and data for indoor scenes (such as NYU/Scannet) are not considered or discussed.
- Qualitative results in Fig 6 and 7 are not very accurate compared to state-of-the-art supervised depth estimation methods, such as MiDAS/DPT. Is there an explanation of why this happens?

---

> ### Author Response · Authors · 2022-11-08
> **Response to YWtU**
>
> Thank you for taking the time to review this publication and for the constructive comments!
>
>  * **Lack of clarity differentiating approaches:** We have reformatted the contents from Sec 5.1 into a table, which should make the contributions and settings of each approach clearer. We will also encourage the reader to refer to the original publications for additional details. We are also planning on releasing the code, which should further clarify the settings required by each method.
>
> * **Self-supervised depth estimation is not well motivated:** We believe that, despite the existing datasets with available ground-truth, self-supervised depth estimation is still a worthwhile research direction to pursue. The primary reason being the ability to scale to larger amounts of data. This is especially relevant in domains in which it is challenging or expensive to collect this ground-truth data.
>
> * **No indoor data is considered:** We agree that this could be an interesting extension to current approaches. However, all benchmarked methods were originally developed and evaluated on the Kitti dataset. Since implementing and testing all of these methods in their current form was already a monumental undertaking, we believe that further adapting to indoor scenarios is beyond the scope of this publication.
>
> * **Qualitative results are inferior to supervised SOTA:** This is expected, since self-supervised problems are much harder to solve than supervised ones. In the case of depth estimation, it is well known that the photometric loss is inaccurate and may not reflect the ground-truth correspondence. Furthermore, purely monocular supervision is still highly sensitive to dynamic objects.
>
> * **Provide additional details on SYNS-Patches:** We have added these details to the updated manuscript. It is worth noting that the depth is clamped to a fixed range (the same used in Kitti) during evaluation.
> Methods using stereo shouldn’t be called self-supervised: The techniques marked as `stereo’ indicate only that the self-supervised photometric error is computed between left and right views, rather than two frames of the same camera. It does not imply that a dedicated stereo reconstruction algorithm has been employed. However, as the reviewer proposed, many methods (all of those marked as D*) use proxy depth maps and regression losses from hand-crafted stereo matching techniques. However, even in the best case scenario (Fused SGBM maps from Depth Hints), these depth maps are not perfectly accurate and contain many errors. As such, they are not comparable to supervised methods using high-quality LiDAR and/or SfM models.
>
> * **Other comments:** We have addressed the remaining comments and suggestions in the revised manuscript.

---

### Review · Reviewer_hVhm · 2022-09-21

**Summary Of Contributions:**

This paper aims to gain a better understanding of recent developments in depth reconstruction methods from two perspectives: (1) by revising evaluation criteria through making changes and corrections datasets and evaluation metric implementations (2) revising the implementations of 16 prior works to fairly include design decisions that improve performance across the board. This paper presents an extensive empirical study to investigate these two directions. The main takeaways are
- When trained using a similar optimization setup and in a unified codebase, all 16 methods get improved performance, and the relative performance improvement over SotA originally reported in the papers is significantly higher than the improvements obtained in the presented study. This indicates that when methods are made more equal, relative improvements are less significant.
- There are flaws in the popular evaluation datasets and their accompanying evaluation code, that when corrected, result in a different ranking of prior methods based on their performance.

This paper also presents a novel dataset (SYNS-patches) of images with accurate ground truth depth from a diverse set of scene types, allowing for . The release of the codebase and the evaluation methodology presented in this paper has potential for high impact in the monocular depth estimation community.


**Broader Impact Concerns:**

I do not have any concerns about the ethical implications of this work.

**Requested Changes:**

**Adjustments that would take a good empirical paper to a great one**
- Can the authors please provide more details about the design of the “Ours” method? Given that a code release will follow with this paper this isn’t so significant, but it would be good to add to the paper.
- While the changes made to the previous methods to make them more comparable do result in improved performance, what were the criteria used to pick which design changes to make to which algorithms? This would also be helpful if added to the paper.

**Extremely Minor**
- Small typo in conclusion: “the change is performance” → “the change in performance”
- Zeki et al. citation needs to be corrected. Please check the correctness of the citations.


**Strengths And Weaknesses:**

**Strengths**

- *Writing*: The paper is overall well written and straightforward to follow, with clear tables and visualizations. The included charts help the reader understand the evidence for the claims made in the paper. The draft is in good shape
- *Identifying the evaluation errors in prior benchmarks (KITTI-Eigen) and determining their impact. Proposing the new KITTI-Eigen-Benchmark*: Sequences of multiple computer vision papers tend to follow the evaluation methods (dataset splits, evaluation scripts) from one popular initial work for the sake of consistency. For practical reasons, comparisons often are done by taking numbers from a previous paper and evaluating the new method using the same evaluation code as before. This paper correctly argues that such an approach leads to errors in evaluation being propagated across many papers. This work follows Uhrig et al. 2019 in identifying the inaccurate ground truth in KITTI , as well as calling out that the ground truth depth is in the LIDAR reference frame rather than the camera reference frame. In Tables 4 and 5 the paper presents evidence of the discrepancy in the ranking of models based on their performance when the KITTI-Eigen is corrected.
- *SYNS patches dataset*: A new testing dataset of images from diverse settings (indoor, outdoor, urban and rural) with high quality ground truth depths to evaluate methods outside of the automotive domain. The paper presents evidence of significantly lower performance on this testing domain than on the automotive KITTI.
- *Thorough comparison of 16 prior depth estimation methods*: This paper presents an evaluation of 16 depth estimation methods using a modern backbone (ConvNeXt), the same optimization setting and per-method changes that increase each baseline’s performance. The main contribution of this evaluation is identifying that relative improvements are significantly lower when the comparison is well designed than what is originally reported.
- *Establishing a new SOTA*: the improved benchmarks through both revising KITTI and the new SYNS-patches dataset as well as the evaluation of previous methods through a unified codebase represents a new picture state of the art in depth estimation. This paper proposes a new method (which appears to be based on the most significant method design decisions identified through the empirical investigation) which ranks highest based on the KITTI-Eigen-Benchmark and SYNS-patches.
- *Identifying that depth regularization hurts performance* - on the fixed KITTI benchmark, this paper identifies that a common technique of smoothness and occlusion-based depth regularization in fact hurts performance, whereas it was likely useful on the original KITTI due to the inaccurate edge boundaries.

**Weaknesses**
- *Clarity of presentation in some cases*: please see more detail in “Requested Changes” below
- *Potential overemphasis on the finding of the effect of different backbones*: While it is valuable to understand the effect of swapping more modern backbones with orders of magnitude more parameters, it is generally understood that doing so results in a "rising tide lifts all boats" kind of effect . I suggest the authors consider revising their claims (2nd to last paragraph in the intro) that swapping a backbone for an old method results in better improvement than changes in algorithm design.

---

> ### Author Response · Authors · 2022-11-08
> **Response to hVhm**
>
> Thank you for taking the time to review this publication and for the constructive comments!
>
> * **Overemphasis on the different backbones:** The main point we are trying to make in that section is the discrepancy between the improved performance reported by original papers compared to that obtained when all models are trained comparatively. In this case, our results show that simply replacing the backbone is a more effective way of improving performance, rather than introducing other complex contributions. We have rephrased these paragraphs to make this clearer.
>
> * **Provide more detail about “Ours”:** As detailed in the response to reviewer YWtU, we have reformatted the contributions and settings of each method as a table, including “Ours”. As the reviewer also points out, we will be releasing the code to train/test all approaches.
>
> * **Criteria used to pick design changes make to which algorithms:** In general, our philosophy was to try to minimise the changes between approaches, with the aim of evaluating only the core contributions of each approach. In practice, this meant establishing a common decoder architecture and retroactively applying contributions that have become standards in the field. This includes the use of the SSIM+L1 loss and edge-aware smoothness regularisations, originally proposed by Monodepth. We also applied the upsampled multi-scale loss proposed by Monodepth2, since that has also been used by all subsequent methods. We have clarified these decisions in the revised manuscript.

---

### Review · Reviewer_WQwG · 2022-10-28

**Summary Of Contributions:**

This paper is about what matters in the architecture of a self-supervised monocular reconstruction. The main finding of the paper is that updating the backbone makes standard baselines outperform recently proposed systems. Moreover, the authors identify fundamental mistakes in the evaluation protocol. They provide code and a new dataset SYNS-Patches.


**Broader Impact Concerns:**

There is no broader impact session needed.

**Requested Changes:**

The results are worth publishing, but if published by themselves, they present a distorted view of reality in monocular depth reconstruction.

The following adjustments are absolutely necessary for publication:

1. Explain and enumerate seen and unseen scenes in the new dataset.
2. Devote a paragraph explaining the weaknesses above:
2a If so many approaches were re-implemented why was no COLMAP used for training/testing?.
2b Why was no prior-free approach tested that extracts video from successive frames on a monocular video.

The above is a good step toward publication.
I would give a strong acceptance if
1. the authors would evaluate a prior-free approach compared to learning based approaches on performance in an unseen environment (not just different parts of the same city in kitti).
2. the authors would add a COLMAP option on one of the best performing approaches. Even a limited evaluation on this would be sufficient.

**Strengths And Weaknesses:**


(+) This is a unique study on the architecture, component, and loss choices.
(+) The paper concentrates on methods that use self-supervision signals like the photo-consistency of frames or feature consistency after warping using the predicted depth and  motion.
(+) The authors re-implemented and reevaluated 16 state of the art methods.
(+) The evaluation was done on real data in urban and real environments
(+) The authors corrected previously identified mistakes in the ground-truth of KITTI-Eigen due to the positioning of Lidar vs camera and the sparsity of the Lidar.
(+) The new dataset contains also two new evaluation measures that consider the distance to depth edges and completeness of depth edges.
(+) The authors found out that ConvNeXt backbone architectures outperform all others.
(+) Evaluation is also done in true 3D comparing depth predictions with point cloud positions.

(-) It would be helpful to always clarify whether we talk about training or inference. For example, in the caption of Fig.~1 "have much higher error as they do not use stereo inputs." It obviously means training since the paper is about mono-depth.


(-) There is no study on the effect of what pose regression method is used.
(-) It is unclear to the reader why the authors do not use COLMAP in training and testing.
(-) This makes it evident that the task of monocular depth reconstruction is ill-defined.  Does monocular mean monocular video or single-frame? If it is self-supervised it has to be video.
(-) If it means monocular video, why do the authors not evaluate any dense visual SLAM systems that do not use priors? One could use a baseline like plane sweeping stereo using the poses obtained by COLMAP.
(-) Why do the authors not evaluate the methods on UNSEEN environments, meaning complete scenes like new cities where a robot has never been before? For example in the VO task, DroidSLAM trains only on TartanAir datasets and tests on EuRoC, ETH-3D, and TUM-RGBD.
(-) There are smaller datasets like NBVC by Mordohai et al. IROS 2020

---

> ### Author Response · Authors · 2022-11-08
> **Response to WQwG**
>
> Thank you for taking the time to review this publication and for the constructive comments!
>
> * **Clarify whether we talk about training or inference:** As the reviewer points out, during inference the depth estimation network only ever makes use of a single frame. As such, any instance in which we mention the use of multiple frames refers to the training procedure. This has been clarified in the reviewed manuscript.
>
> * **No study on the effect of pose regression:** All approaches trained with monocular video make use of the same pose regression system. As detailed in Appendix A.2, we use a CNN to predict the relative pose (in axis-angle format) between two consecutive video frames concatenated channel-wise. Furthermore, in this paper we are focused on the task of single-view depth prediction. As such, we use pose estimation only as a surrogate task to compute the photometric view synthesis losses during training. We have added a paragraph to the revised manuscript explaining this.
>
> * **Why not use COLMAP during training and testing?** As discussed, this paper focuses on monocular depth estimation from a single frame at test time, not on structure-from-motion or SLAM. Regardless, even at train time COLMAP is not suited towards reconstructing environments such as those in the Kitti dataset. This is due to the sequential nature of the data, which doesn’t contain viewpoint variation, and the presence of dynamic vehicles, which are reconstructed incorrectly. We ran COLMAP on multiple Kitti sequences, all leading to poor results. In several cases, the model had not converged after 8+ hours. In other cases, the quality of the reconstruction was suboptimal, with miss-registered images and incorrect depth estimates for dynamic objects.
>
> * **The task of monocular depth reconstruction is ill-defined. Does it mean monocular video or single-frame?** During both training and inference, we focus on depth prediction from a single frame. In other words, the depth estimation CNN only requires a single RGB image as its input. However, to train this network, we use additional support frames to perform view synthesis. These frames can either come from a stereo pair or from a monocular video. In the case where we use monocular video, we additionally train a CNN to regress the pose between consecutive frames. This pose prediction network is only used to compute the photometric view synthesis losses during training, and is not required during inference.
>
> * **Why not evaluate dense SLAM systems?** At inference time, the systems in our paper need only a single image to produce depth, while SLAM and SfM require a whole sequence. This paper focuses on single-image depth prediction, not on reconstruction from a temporal sequence. As such, we consider SLAM and SfM systems as a separate task that is out of the scope of this paper.
>
> * **Why not evaluate on unseen environments, i.e. different cities?** The evaluation done on SYNS-Patches consists of an entirely new unseen environment. As detailed in Sec 5.4, we evaluate the same models trained on Kitti, without re-training or finetuning on SYNS-Patches. We will further clarify this in the paper. The remaining datasets proposed are mainly indoor datasets to evaluate SLAM systems. Once again, we do not provide an evaluation of the pose estimation network because it is not needed at inference time. We only use it as a surrogate task during training.

---

### Review · Reviewer_QF4o · 2022-11-16

**Summary Of Contributions:**

This paper re-evaluates existing self-supervised monocular depth estimation networks and investigates the significance of some designs. To this end, the authors re-implement several representative methods and evaluate them both on the original benchmarks and the proposed SYNS-Patches test sets. The authors conclude that the backbone network is critical for improved performance, while some other designs, such as the regularization loss, are less significant or even decrease the performance.

**Broader Impact Concerns:**

I do not see a potential ethical liability issue in the paper.

**Requested Changes:**

- Depth regularization: the authors should dive deeper on the depth regulariziation.
  - First, nonlocal reuglarization term (or at least bileratal term) should be used as the edge-aware regularziaton. The gradient weighted one (Godard et al., 2017) is too sensitive to noises.
  - Second, if remove depth regularization is better, the author should investigate how the network avoid ill-posed solutions. For example, what are the addtional sources that provide the regularziation, e.g., the network architecture.

- Pose: the authors should use "ground-truth" poses for monocular sequences during training. It will exclude inaccurate poses and refocus to the depth network design.

- Datasets: the authors should include less layout biased dataset, both indoor datasets such as ScanNet (training & testing) and outdoor datasets such as DIODE (testing only) into evaluation.

**Strengths And Weaknesses:**

Strengths
+ The authors re-implemented and re-evaluated many representative methods in a unified codebase, which is fair and less biased.
+ The authors proposed a new test benchmark that contains more diverse scenes than KITTI.
+ The authors also corrected errors in KITTI benchmarks during evaluation.

Weaknesses
- The conclusion is somewhat trivial, improving backbone networks
- The authors observe that the depth regularization decreases the performance, but lacks a deeper analysis.
- For the monocular sequence, the the pose is estimated from a network, which makes it unclear whether the decreased/increased performance is from other design choices for better depth.
- In terms of the existing datasets, the authors only investigate KITTI, which has a strong layout bias and may limit the generality of the conclusion.

---

> ### Author Response · Authors · 2022-11-22
> **Response to QF40**
>
> Thank you for taking the time to review this publication and for the constructive comments!
> * **The conclusion is somewhat trivial, improving backbone networks:** We would argue that the conclusion is much more significant than that. Our conclusion is that the evaluation protocol used by the field is not fit for purpose. Our analysis shows that over the years, significant gains in performance have been misattributed to numerous complex developments, but were in fact due to simple improvements in design choices. Our solutions to this problem is to introduce a new and fair evaluation framework to ensure that novel contributions can be compared in a consistent manner.
> * **Non-local or bilateral regularisation should be used as edge-aware regularisation:** We agree that these novel regularisation concepts would be interesting to test. However, the focus of this paper is to re-evaluate existing and well-established contributions to the field. We focus on edge-aware regularisation because it has been used by almost every method since its introduction. Furthermore, we indeed show that, when correcting the ground-truth used for evaluation, this regularisation loses its effectiveness.
> * **If removing the depth regularisation is better, authors should investigate how the network avoids ill-posed solutions:** Our intuition is that the largely overlapping receptive fields in neighbouring points implicitly allows the network to produce a smooth output, especially in the case of textureless regions. As such, the improved performance of edge-aware regularisation was simply due to oversmoothed predictions combined the incorrect KITTI Eigen ground-truth, which contains background bleeding at object boundaries.  We will add this discussion to the final manuscript.
> * **With monocular supervision the pose is estimated by a network, so it is unclear whether the change in performance is due to inaccurate poses:** This would be an interesting ablation to test. We are happy to include this if the reviewer believes it is important. However, since all methods use the same pose estimation framework, we believe that the effect of inaccurate poses will be constant across all approaches.
> * **Authors should include additional datasets such as ScanNet (indoor) and DIODE (outdoor):** We agree with the sentiment that prior work has been restricted to too narrow a domain. However, we believe this paper already addresses this concern, since one of the core components of the evaluation procedure is the novel SYNS-Patches dataset. This represents an evaluation dataset with a much wider variety of natural, urban and outdoor scenes. Meanwhile, the outdoor scenes from DIODE only contain urban settings. As detailed in the response to YWtU, all evaluated contributions were designed for outdoor settings. Adapting all methods to training in indoor scenarios is beyond the scope of this publication.

---

### Author Response · Authors · 2022-11-08
**Updated Revision**

Thank you all reviewers for your comments. Based on these, we have provided an updated revision of the paper with the following list of changes.

* Rephrased backbone improvement claims in Sec 1 par. 4. (hVhm)
* Clarified use of stereo pairs during training in Fig 1. (WQwG)
* Added dataset depth distribution and densities to Sec 3.2. (YWtU)
* Clarified SYNS-Patches as an unseen testing dataset in Secs 3.2 (final par.) and 5.4 (first par.). (WQwG)
* Added discussion on pose prediction network to Sec 4, par. 2. (WQwG)
* Added frames-per-second to backbone ablation Tab 2. (YWtU)
* Clarified use of hand-crafted disparity maps as proxy supervision in Sec 5.1. (YWtU)
* Added common design decisions discussion to Sec 5.1, par. 2. (hVhm)
* Formatted method settings and contributions as table in Sec 5.1. (YWtU, hVhm)
* Added paragraph discussing generalization and depth discontinuities to Sec 6 par. 2. (YWtU)
* Corrected minor typos and comments.
     (All)

---

### Decision · Action_Editors · 2022-12-15

**Recommendation:** Accept with minor revision

**Comment:**

3 out of 4 reviewers are positive about the paper and suggest acceptance. The more negative reviewer would have liked to see more base architectures (e.g. transformer-based) and more analysis into the loss function. I think the paper as is has already enough quality and value for acceptance, but suggest a minor revision where the authors can incorporate the additional feedback provided in reviewer YWtU decision, and if possible also the loss function analysis requested by reviewer QF4o.

Congratulations!

**Audience:**

Self-supervised monocular depth estimation is a problem that touches different strands of research -- geometry, self-supervised learning, graphics. Additionally, monocular depth estimation is of interest to fields as varied as self-driving cars and text-to-image generation, so I believe there will a sufficient number of people interested.

**Claims And Evidence:**

All reviewers agree on the quality of the evaluation reported in this paper. In fact it fixes issues in an existing popular benchmark and introduces a new one. It performs an extensive evaluation and analysis of 16 different methods from the literature leading to new SOTA results.